# Microfluidic manipulation by spiral hollow-fibre actuators

Sitong Li[1], Rui Zhang [2], Guanghao Zhang[1], Luyizheng Shuai[1], Wang Chang[1], Xiaoyu Hu[1], Min Zou[1], Xiang Zhou[1], Baigang An[3], Dong Qian[2] & Zunfeng Liu [1✉]

A microfluidic manipulation system that can sense a liquid and control its flow is highly desirable. However, conventional sensors and motors have difficulty fitting the limited space in microfluidic devices; moreover, fast sensing and actuation are required because of the fast liquid flow in the hollow fibre. In this study, fast torsional and tensile actuators were developed using hollow fibres employing spiral nonlinear stress, which can sense the fluid temperature and sort the fluid into the desired vessels. The fluid-driven actuation exhibited a highly increased response speed (27 times as fast as that of air-driven actuation) and increased power density (90 times that of an air-driven solid fibre actuator). A 0.5 K fluid temperature fluctuation produced a 20° rotation of the hollow fibre. These high performances originated from increments in both heat transfer and the average bias angle, which was understood through theoretical analysis. This work provides a new design strategy for intelligent microfluidics and inspiration for soft robots and smart devices for biological, optical, or magnetic applications.

[1] State Key Laboratory of Medicinal Chemical Biology, College of Chemistry and College of Pharmacy, Key Laboratory of Functional Polymer Materials, Frontiers Science Center for New Organic Matter, Nankai University, Tianjin 300071, China. [2] Department of Mechanical Engineering, University of Texas at Dallas, Richardson, TX 75080, USA. [3] School of Chemical Engineering, University of Science and Technology Liaoning, Anshan 114051, China. ✉email: liuzunfeng@nankai.edu.cn

nspired by biofluidic systems, microfluidics have been developed and applied in various fields, such as in devices[1,2], biotechnology[3–9], medicine[10,11], and electronics[12,13]. Microfluidic manipulation enables precise delivery of small amounts of liquid to the target position for sensing or analysis[14], where the transported fluid is precisely sensed and controlled (including volume, temperature, and concentration) to meet the required conditions[15–22]. One challenge is to obtain real-time data of the small amount of fast-flowing liquid in a hollow fibre because conventional sensors have difficulty fitting the small space in a microfluidic device. In addition, fast sensing and actuation are not easy because of the fast flow of the fluid. Therefore, a compact microfluidic manipulation system that provides *real-time* sensing and fast actuation is highly desirable.

In recent years, stimuli-responsive materials have been incorporated into micrometre channels to realise microfluidic control[23,24]. Typical materials include magneto- or electro-active materials[25], hydrogels[26–28], and liquid crystals[29–31], which are controlled by magnetic[25] or electric fields[32,33], temperature[34,35], pH[36], and light[37,38]. Magnetic, electrical, and optical stimuli require additional equipment[39]; in addition, these responsive materials require complicated preparation procedures. Therefore, development of a microfluidic manipulation technique that provides *real-time* sensing of the transported fluid and fast actuation employing existing commercial tubing materials is highly desirable. Moreover, for practical applications, the response time, actuation force, and actuation stroke still need to be improved.

Twisted fibres can generate torsional and tensile actuation by volume expansion, and various fibre actuators have been developed[40–56]. In the present study, we envisage that inserting a twist into a hollow fibre may generate a spiral hollow-fibre actuator. The actuation mechanism of the twisted hollow fibre is similar to that of a twisted solid fibre. When the fibre sheath is heated by the flowing hot liquid, the sheath material contracts in the axial direction and expands in the radial direction, causing torsional and tensile actuation for a twisted and a coiled hollow-fibre actuator, respectively. Consequently, the actuation of the hollow fibre in response to a liquid with different temperatures would result in microfluidic position control. However, to date, there have been no reports on the incorporation of twist insertion into hollow fibres for actuators. This difficulty may be caused by the buckling and collapse of the hollow fibre during twist insertion, which blocks liquid flow.

Liquid transport in the hollow fibre would result in efficient heat transfer at the fluid-hollow fibre interface, and consequently, a faster response would be expected. Furthermore, the outer surface of a fibre always exhibits a higher bias angle than the inner fibre surface[57], and consequently, increased actuation stress and work output are expected for a hollow fibre with the same mass but a larger outer radius (Supplementary Note 3).

In this study, a multifunctional microfluidic system has been developed to control the outflow position by sensing the temperature of the transported liquid (Fig. 1a, b). This is realised by inserting a twist into a hollow fibre, which was initially plugged with a metal wire to avoid twist-induced collapse, followed by thermal annealing to set the shape. Rotation, contraction, and extension are realised for twisted, homochiral, and heterochiral coil hollow-fibre actuators, respectively, by flowing hot liquid. The fluid-driven actuation is 27 times as fast as that driven by airflow; the work capacity and power density are 1.5 times and 90 times, respectively, those of a solid fibre. The torsional hollow-fibre actuator exhibits a high sensitivity of a 20° rotation after a 0.5 K fluid temperature change. The current strategy is applicable to common hollow-fibre materials and is highly stable for 10,000 investigated heating-cooling cycles.

## Torsional hollow-fibre actuators

The hollow-fibre actuator was prepared using low-density polyethylene hollow fibres (PEHFs) with different inner (x, in μm) and outer (y, in μm) diameters, which were denoted as $PEHF_{x-y}$. Upon heating, the PEHF expands in the radial direction and contracts in the length direction[58]. This anisotropic thermal expansion behaviour is ascribed to the taut tie molecular morphology formed during the fabrication of the PEHF[59–61]. The PEHF exhibited a breaking strength of 39.4 MPa and a fracture strain of 188.4% (Supplementary Fig. 1a), indicating reasonably high strength and flexibility. The aforementioned characteristics enable the PEHF to be an ideal candidate to prepare actuators if a twist can be inserted into the PEHF.

Because the hollow fibre is easily buckled by direct twist insertion, a thin copper wire was inserted into the hollow fibre to avoid collapse. By inserting a copper wire slightly thinner than the hole of the PEHF, the PEHF did not collapse when inserting up to 400 turns $m^{-1}$ twist. The twisted PEHF with copper wire was then annealed at 108 °C for 1 h with both ends tethered to thermally set the shape. This annealing temperature is slightly less than the melting temperature of the PEHF (111 °C, Supplementary Fig. 1b), which was slightly softened at this temperature. After cooling to room temperature (25 °C) and removing the tethering, ~ 3% twist loss was observed (Supplementary Fig. 12). For the convenience of expression, in the following context, the initially inserted twist was used to calculate the twist density without subtracting the twist loss. The copper wire was thereafter removed, and the twisted PEHF did not collapse and was only slightly deformed (Fig. 1d). Successful twist insertion into the hollow fibre provided the possibility of preparing torsional and tensile actuators if volume expansion occurred.

The twisted fibre would undergo torsional rotation if the fibre experiences volume expansion[62]. Therefore, rotation was expected when the twisted PEHF was heated (Fig. 2a). Despite inserting a twist into the polymer fibre, the highly oriented semicrystalline form is also key to the actuation capability of the fibre. Two-dimensional wide-angle X-ray scattering (2D WAXS) experiments were carried out, and the results show that the fibres are in highly oriented semicrystalline form (Supplementary Note 1 and Supplementary Fig. 1d). In our experiments, torsional actuation was observed by flowing hot water in a twisted $PEHF_{580-990}$. To facilitate measurement of the rotation angle, a paddle (with a mass of 1/50 that of the hollow fibre) was taped at the outlet of the $PEHF_{580-990}$ (Fig. 2b). A thermal camera was used to monitor the surface temperature of the hollow fibres during actuation. If not specified, water at different temperatures was pumped into the hollow fibre at a flow rate of 0.5 g $s^{-1}$. The inserted twist density and rotation angle were normalised to the nontwisted length of the hollow fibre. The hollow-fibre actuators were trained for 3 cycles to obtain repetitive actuation performance.

A $PEHF_{580-990}$ actuator with an inserted twist density of 400 turns $m^{-1}$ rotated by 189° $cm^{-1}$ in 1.7 s when flowing water at 95 °C, corresponding to a rotation speed of 44 rpm (Fig. 2c). Thereafter, the actuator completely rotated back in 1.6 s when flowing water at 25 °C. This response time (1.7 s) is approximately 1/13 that for hot air flow (22.8 s) (Fig. 2d). The rotation angle and rotation speed were monotonically correlated with the water temperature and inserted twist density (Fig. 2e, f). The torsional rotation of the hollow-fibre actuator induced by flowing hot water indicates the possibility of torsional manipulation of the microfluidics through sensing of the water temperature. Moreover, the manipulation can be delicately tuned by changing the twist density.

We thereafter compared the torsional actuation performance of the hollow-fibre actuator with those in previous studies. As PEHFs with different outer diameters were used, we attempted to

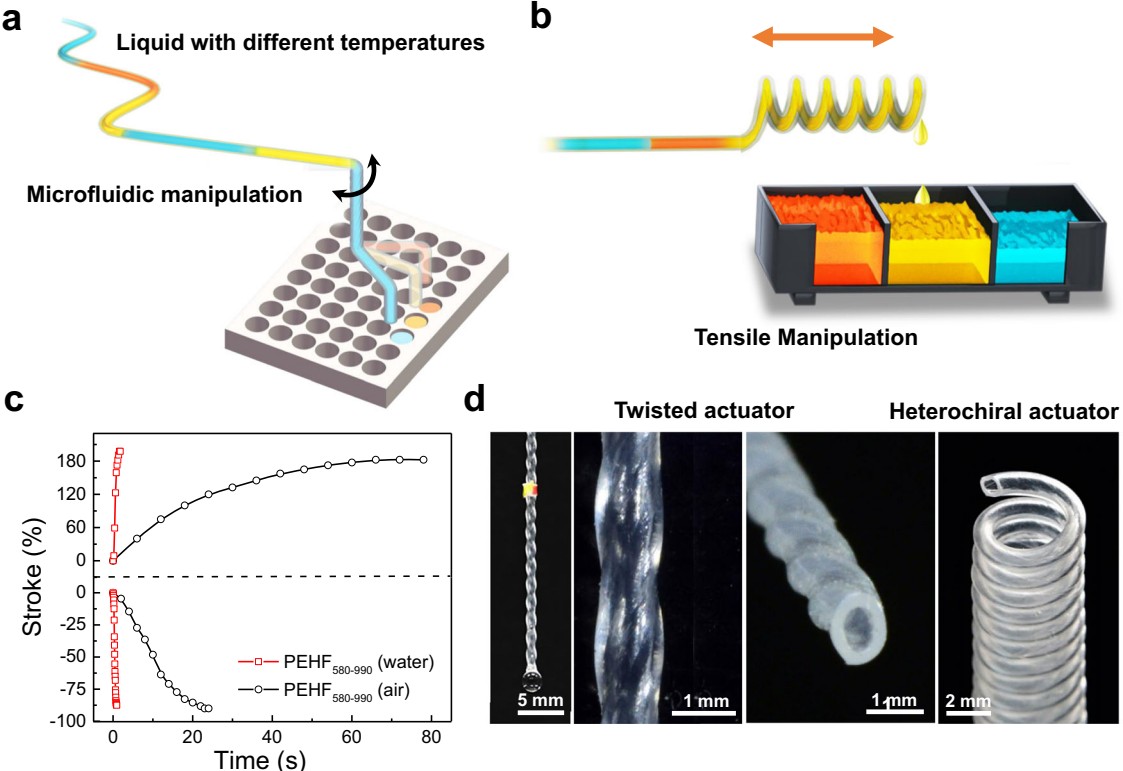

**Fig. 1 Hollow-fibre actuators for microfluidic manipulation. a** Schematic of a compact microfluidic manipulation device that can sense the liquid temperature and control the liquid flow by employing a twisted hollow-fibre actuator. **b** Schematic of the tensile hollow-fibre actuator used for sensing the liquid temperature and sorting the liquid into the desired cell. **c** Comparison of the heterochiral and homochiral PEHF$_{580\text{-}990}$ actuators driven by 95 °C water and air. **d** Images of the coiled and twisted hollow-fibre actuators.

normalise the rotation angle by the outer diameter. For a certain surface bias angle (α), the twist density ($T$) is inversely proportional to the fibre diameter ($r$) following the equation α = $\tan^{-1}(2\pi r T)$[63]. Therefore, a thinner fibre would rotate more than a thicker fibre for the same change in bias angle. Consequently, we multiplied the rotation angle by the outer diameter of the hollow-fibre actuator to obtain a normalised value. The 189° cm$^{-1}$ rotation angle for the PEHF$_{580\text{-}990}$ actuator corresponds to a normalised rotation angle of 18.9°. This is comparable to that of the torsional actuators reported in previous studies, such as carbon nanotube yarn (7.12°)[63], graphene fibre (38.3°)[64], silk yarn (5.2°)[47], cotton yarn (0.85°)[42], bamboo fibre (14.7°)[65], and nylon 6 fibres (4.5°)[58]. Because a higher inserted twist density resulted in a larger bias angle, we also normalised the torsional rotation by dividing it by the twist density. The 189° cm$^{-1}$ rotation angle for the PEHF$_{580\text{-}990}$ actuator with an inserted twist density of 400 turns m$^{-1}$ corresponds to a 13.1% change in inserted twist. This is also comparable to that of the torsional actuators reported in previous studies, such as carbon nanotube yarn (1.16%)[63], graphene fibre (6.5%)[64], silk yarn (10%)[47], cotton yarn (9.8%)[42], and bamboo fibre (22%)[65].

**Tensile hollow-fibre actuators**
Next, we investigated whether tensile actuation could be achieved by employing spiral hollow fibres. The twisted PEHF with a copper wire was wrapped around a mandrel to form a coil and annealed for 1 h at 108 °C to thermally set the shape, followed by cooling to room temperature and removal of the copper wire. The coil geometry converts the rotation actuation into a spiral distance change.

Coiled hollow-fibre actuators exhibit different chiralities. The same coiling and twist directions produce a homochiral coil,

which would contract upon volume expansion, and opposite coiling and twist directions produce a heterochiral coil, which would expand upon volume expansion (Supplementary Fig. 11)[45]. The coil geometry, including the pitch distance, coil length, and spring index, significantly affects the actuation performance. The pitch distance is the distance between two neighbouring coils. To obtain a large actuation stroke, the pitch distance of the homochiral actuator needs to be large to allow the coil to contract, whereas that of the heterochiral actuator needs to be small to allow the coil to extend. A larger mandrel diameter always results in a shorter coil length for the same pitch distance. Consequently, we define the ratio of the coil diameter to the outer diameter of the hollow fibre as the spring index ($C$) to describe the actuation stroke.

We thereafter investigated the tensile actuation performance using a homochiral PEHF$_{580\text{-}990}$ actuator with an inserted twist density of 300 turns m$^{-1}$ and a spring index of 4.0. The actuator with a load of 107 kPa contracted by 50% in 1.2 s when flowing 90 °C water at a flow rate of 1.72 g s$^{-1}$ (Supplementary Movie 1). The nonloaded actuator with an inserted twist density of 200 turns m$^{-1}$ demonstrated an even higher contractile stroke (87.5%, until coil-coil contact) within a shorter time (0.88 s), which is 1/27 of the response time obtained by using air heating (24 s) (Fig. 1c). Flowing 25 °C water at a flow rate of 1.72 g s$^{-1}$ resulted in fast recovery to the initial length in 0.76 s (Fig. 3a, inset). No decay was observed in the tensile stroke and actuation speed for 10,000 cycles of actuation driven by consecutively flowing hot (90 °C) and cold water (25 °C). The actuation stroke and response time were monotonically correlated with the water temperature (Supplementary Fig. 3a, b). Owing to heat dissipation into the air from the hollow fibre surface, the outlet temperature decreased from 94.5 to 87.5 °C as the water flow rate

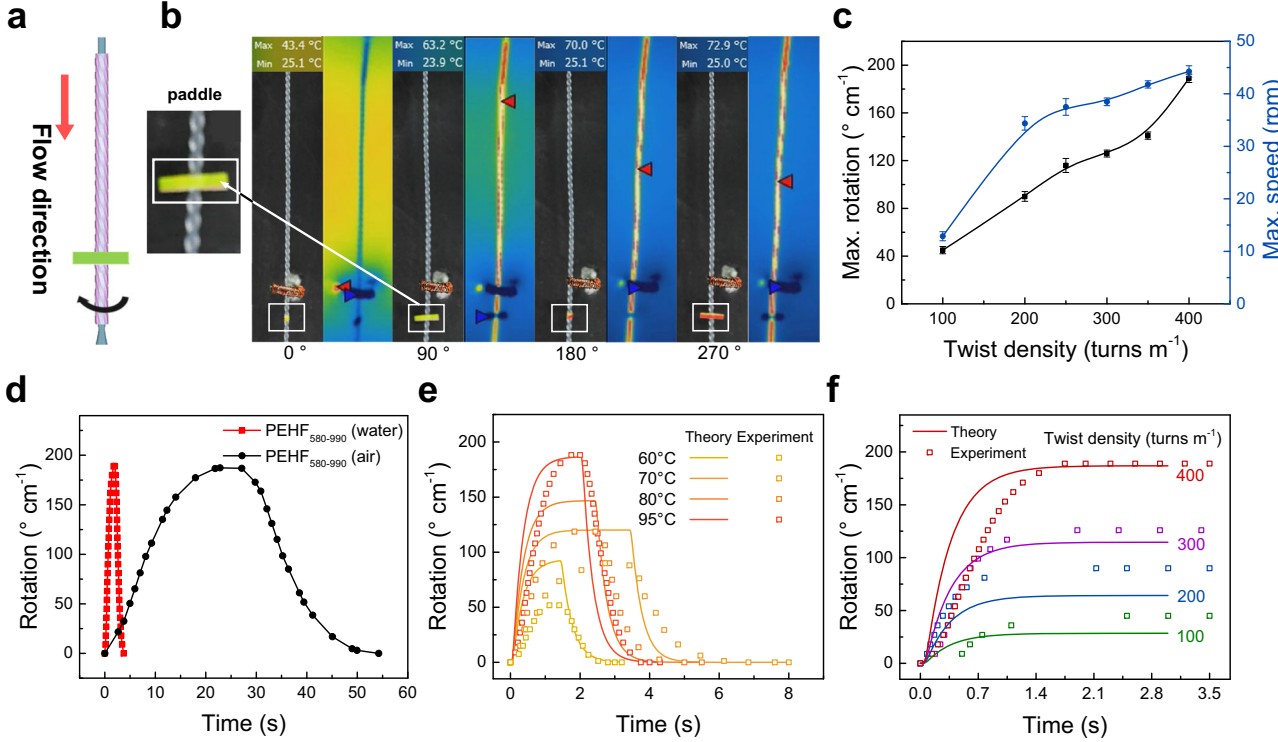

**Fig. 2 Actuation performance of torsional PEHF$_{580-990}$ actuators. a** Schematic and **b** serial images of a twisted PEHF$_{580-990}$ actuator during water flow. **c** Maximum rotation angle and maximum rotation speed as a function of inserted twist density. **d** Comparison of the torsional PEHF$_{580-990}$ actuator driven by 95 °C water and air. **e**, **f** Experimental and theoretical results of the rotation angle as a function of time for the twisted PEHF$_{580-990}$ actuator at different temperatures (**e**) and inserted twist densities (**f**). If not specified, in the following figure captions, the actuation was driven by 95 °C water, and the water flow rate was 0.5 g s$^{-1}$. If not specified, the error bars in **c** and in the following figures indicate standard deviations.

decreased from 0.5 to 0.25 g s$^{-1}$, and simultaneously, the response time of the actuator increased from 1.7 to 2.8 s (Supplementary Fig. 3a). As flowing 80 °C water in the homochiral PEHF actuator already resulted in coil-coil contact, the actuation stroke was maintained at a maximum value of 87.5% as the water flow rate decreased from 0.5 to 0.25 g s$^{-1}$ (Supplementary Fig. 3b).

As different sheath thicknesses of hollow fibres is generally used in microfluidics, in this section, we investigated the actuation performance as a function of the sheath thickness. Homochiral hollow-fibre actuators with different sheath thicknesses were prepared, which had the same surface bias angle (31.9°), pitch distance (9 mm), and spring index (4.0). Three types of PEHFs were used in this study: PEHF$_{380-1090}$, PEHF$_{580-990}$, and PEHF$_{280-640}$, corresponding to sheath thicknesses of 710, 410, and 360 μm, respectively. Figure 3b and Supplementary Fig. 6a, b show images of these coiled hollow-fibre actuators during flowing of 95 °C water at a flow rate of 0.366 g s$^{-1}$ at different time. The decrease in the sheath thickness from 710 to 360 μm resulted in faster heat transfer, and the response time decreased from 3.2 to 1.0 s (Supplementary Fig. 6c). All the samples can form coil-coil contact, and there was a negligible difference in the actuation stroke for these samples with different sheath thicknesses. The fast response and large actuation stroke of the coiled hollow-fibre actuators when the temperature of the transported fluid changes allow precise sensing and microfluidic manipulation. For example, the coiled hollow-fibre actuator would spontaneously change its length with the change in the flowing liquid temperature, thereby indicating the real-time temperature of the fluid; in addition, the large actuation stroke of the hollow fibre in response to the fluid temperature change would

provide the possibility to sort the liquid with different temperatures into desired vessels. Such a fluid-driven actuator can be employed as a new type of artificial muscle with fast and large stroke in soft robots, manipulators in production lines, exoskeletons, and wearable power assist robots.

We thereafter investigated the factors affecting the actuation performance, including the actuation temperature, spring index, and twist density, for both homochiral and heterochiral PEHF actuators. As the temperature of the flowing water increased from 60 to 95 °C, the contractile stroke increased from 5.4% to 56.2% for the homochiral PEHF$_{580-990}$ actuator with a spring index of 4.0 and an inserted twist density of 100 turns m$^{-1}$ (Supplementary Fig. 7a). As the inserted twist density increased from 50 to 250 turns m$^{-1}$, the contractile stroke increased from 12.3% to 89.6% for the homochiral PEHF$_{580-990}$ actuator when flowing water at 95 °C (Fig. 3c). Similarly, the heterochiral actuators (Fig. 3d) exhibited an increased elongation stroke with increasing water temperature and twist density. Increasing the spring index resulted in increased actuation stroke until complete coil-coil contact for the homochiral PEHF$_{580-990}$ actuator with a pitch distance of 9 mm (Supplementary Fig. 7b). The decreased actuation stroke after coil-coil contact for larger-spring-index coils can be ascribed to the decrease in the initial coil length before actuation.

We hypothesize that the actuation mechanism in the proposed hollow-fibre actuator is different from that of a McKibben muscle, which is hydraulically driven. To test this hypothesis, we added an experiment to demonstrate the effect of water flow on the actuation performance. No actuation occurred when room-temperature fluid flowed through the twisted hollow fibre (Supplementary Figs. 3c and 4). Actuation occurred when a hot fluid flowed through the twisted hollow

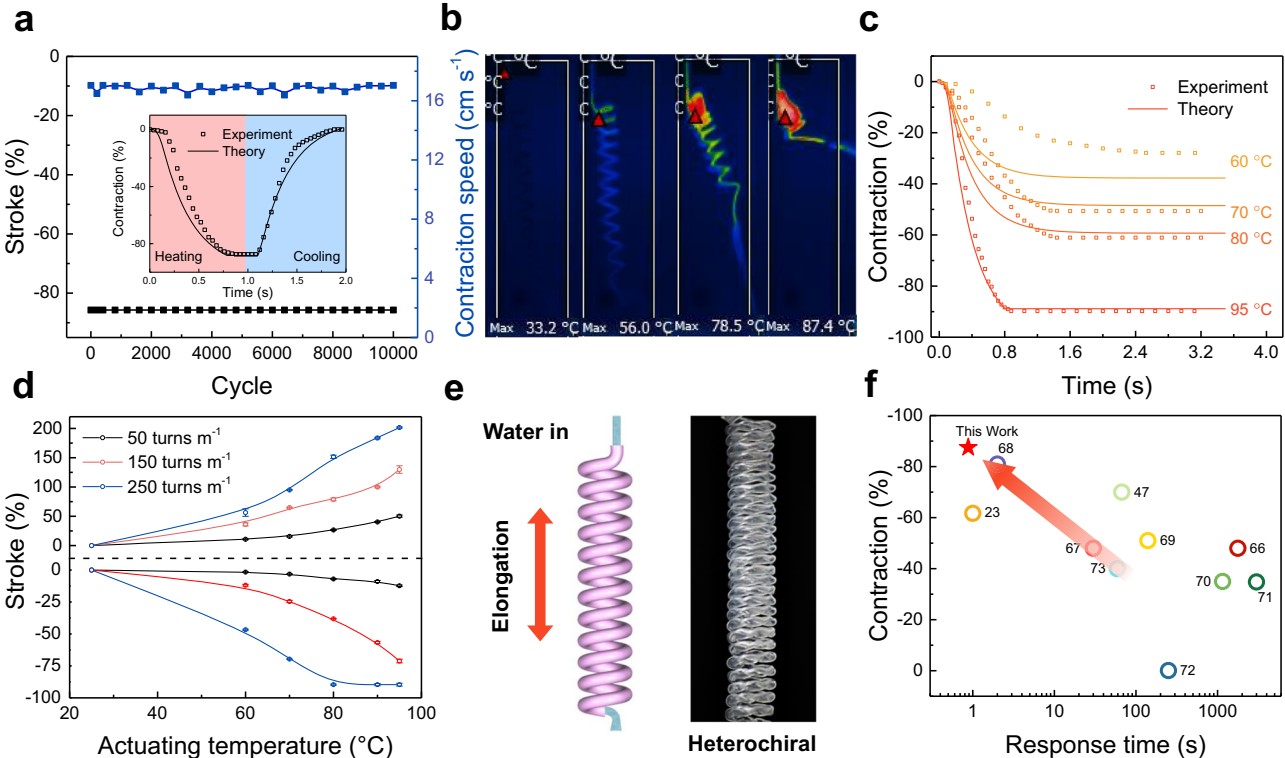

**Fig. 3 Actuation performance of load-free tensile hollow-fibre actuators. a** Tensile stroke and contraction speed during 10,000 fluid-driven heating-cooling actuation cycles. Inset: time dependence of contraction driven by alternatively flowing 90 °C and 25 °C water at a flow rate of 1.72 g s$^{-1}$. **b** Infrared images of the homochiral PEHF$_{580-990}$ actuators during water flow. **c** Experimental and theoretical results of a homochiral PEHF$_{580-990}$ actuator driven by flowing water at different temperatures. **d** Actuation stroke as a function of water temperature for heterochiral and homochiral PEHF$_{580-990}$ actuators with different inserted twist densities. **e** Schematic illustration and photograph of the heterochiral PEHF$_{580-990}$ actuator. **f** Comparison of the contraction and response time of PEHF$_{580-990}$ actuators with those of typical polymer fibre actuators reported in previous studies.

fibre, and the actuation stroke and actuation speed increased with increasing fluid temperature (Figs. 2e and 3c). These results show that the fluid-driven hollow-fibre actuator is hydrothermally driven, not hydraulically driven.

Next, we investigated the ability of the hollow-fibre actuator to carry a load (Fig. 4a), including the actuation stroke, actuation stress, specific work capacity, and cycling stability. For the convenience of expression, the isobaric stress was calculated as the weight of the load divided by the cross-sectional area of the sheath of the hollow fibre according to previously published papers[49,58,63]. For better understanding, we also provide figures directly using the mass of the load in place of the stress in the supporting information (Fig. 4b, c and Supplementary Fig. 8).

A homochiral PEHF$_{580-990}$ actuator, with a spring index of 4.0 and a pitch distance of 9 mm, was actuated by consecutively flowing water at 90 °C and 25 °C, achieving good repetitive actuation performance at different loading stresses (Fig. 4b and Supplementary Fig. 8a). By increasing the loading stress, the contractile stroke monotonically decreased, and the specific work output increased to a plateau of 0.25 MPa (Supplementary Fig. 8b). The specific work capacity increased with increasing twist density (Fig. 4c). As the bias angle at the outer surface is higher than that at the inner surface, we investigated whether hollow-fibre actuators with a higher sheath ratio realised higher work output, where the sheath ratio was defined as the sheath thickness divided by the outer diameter of the sheath. The work capacity increased with decreasing sheath ratio for a wide range of loading stresses (Fig. 4d). The power density of the PEHF$_{580-990}$ actuator was ~ 90 times (46.5 W kg$^{-1}$) that of an air-driven solid PE fibre (0.52 W kg$^{-1}$) (Supplementary Fig. 9), and the self-coiled

nylon 6 hollow-fibre actuator exhibited a specific work capacity (1.72 J g$^{-1}$) 1.5 times that of the solid nylon fibre actuator (1.13 J g$^{-1}$) with the same diameter (Fig. 4e). The combination of the actuation stroke (87.5%) and response time (0.88 s) is thus far the best among the polymer fibre actuators reported in previous studies[23,47,66–73] (Fig. 3f), and the mass-normalised specific work capacity generated per Kelvin is also the best compared to the thermally driven individual polymer fibre actuators[58,66,69,71,73–77] (Fig. 4f). Such high-performance parameters are a result of the combination of efficient heat transfer during fluid flow and the high average bias angle in the hollow fibre sheath.

## Thermal-mechanical modelling of hollow-fibre actuators

As the hollow-fibre actuator was driven by flowing hot liquid, theoretically correlating the actuation behaviour with heat transfer is highly desirable to understand the actuation mechanism for design of hollow-fibre actuators for precise microfluidic manipulation. Therefore, we performed thermal-mechanical modelling of torsional and tensile hollow-fibre actuators by quantifying the heat transfer between the flowing liquid and hollow fibre (Supplementary Note 2). The temperature change of the hollow-fibre actuator is contributed by the forced convection of the flowing liquid (internal) and the natural convection of the ambient (external) air. To study this transient heat conduction problem, we established a two-step model. First, we modelled the radial heat conduction at an arbitrary cross-section of the hollow-fibre actuator by the following governing equations:

$$\frac{\partial^2 \theta}{\partial r^2} + \frac{1}{r}\frac{\partial \theta}{\partial r} = \frac{1}{\alpha}\frac{\partial \theta}{\partial t} \ in \ R_i \leq r \leq R_o, \tag{1a}$$

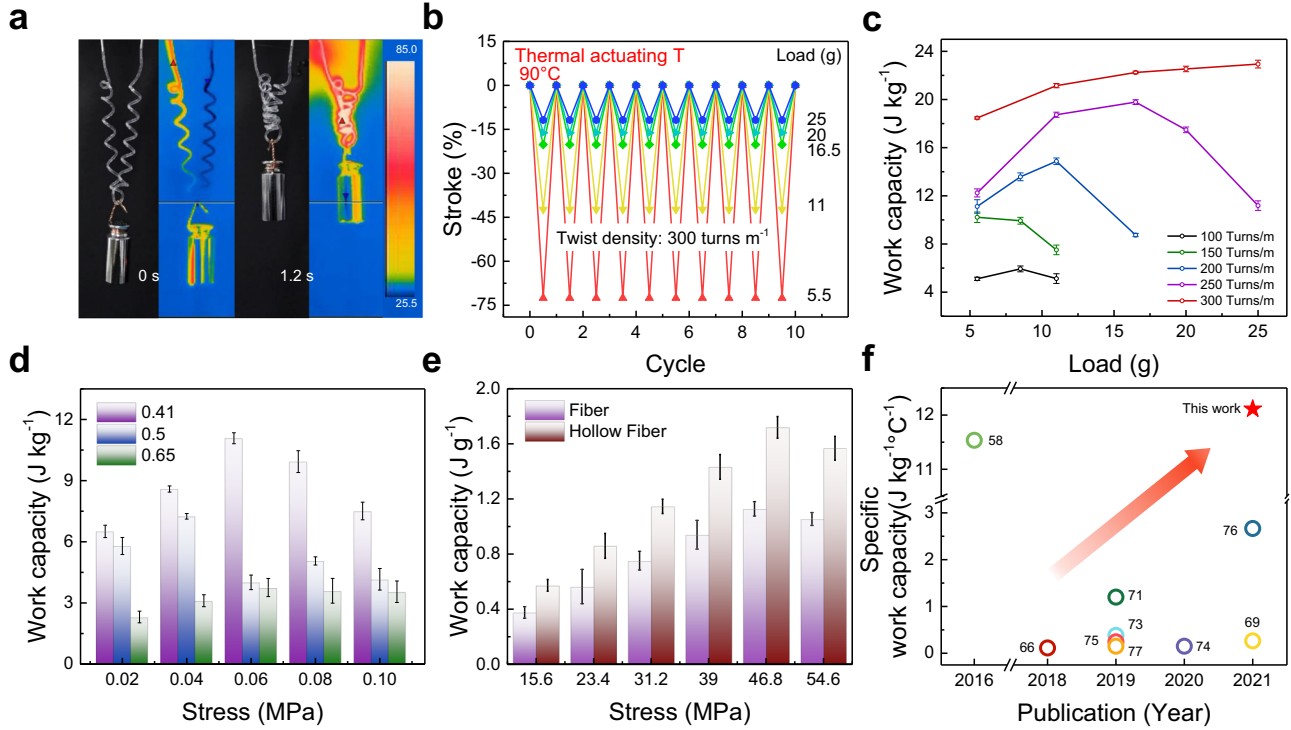

**Fig. 4 Actuation performance of fluid-driven homochiral hollow-fibre actuators. a** Optical and infrared images of a PEHF$_{580-990}$ actuator lifting a 10-g load in 1.2 s when flowing 90 °C water at a flow rate of 1.72 g s$^{-1}$. **b** Maximum contraction at different masses of the load for 10 consecutive heating-cooling cycles of homochiral PEHF$_{580-990}$ actuators with a spring index of 4.0 and an inserted twist of 300 turns m$^{-1}$ driven by alternatively flowing 90 °C and 25 °C water. **c** Work capacity of PEHF$_{580-990}$ actuators with different inserted twist densities and applied loads. **d** Work capacity of PEHF actuators with different sheath ratios at different loads. **e** Comparison of the work capacity for nylon 6 actuators made of hollow and solid fibres at different loads. The isobaric stress in **d** and **e** was calculated as the weight of the load divided by the cross-sectional area of the hollow fibre sheath. **f** Comparison of the specific work capacity of nylon hollow fibres in this work with that of single-filament pure-polymer fibre actuators in previous studies. The specific work capacity is defined as $E/(m \cdot \Delta\theta)$, where $E$ denotes the work output, $m$ denotes the mass, and $\Delta\theta$ is the temperature change.

$$k\frac{\partial\theta}{\partial r} = h_w(\theta - \theta_w) \ at \ r = R_i, \tag{1b}$$

$$-k\frac{\partial\theta}{\partial r} = h_a(\theta - \theta_a) \ at \ r = R_o, \tag{1c}$$

$$\theta = \theta_0 \ at \ t = 0, \tag{1d}$$

where temperature $\theta$ is a function of the radial position $r$ and time $t$, and the inner and outer radii of the hollow fibre are denoted by $R_i$ and $R_o$, respectively. Thermal diffusivity $\alpha = k/\rho C_p$ is a material constant that depends on thermal conductivity $k$, mass density $\rho$, and specific heat capacity $C_p$. The convective heat transfer coefficients of water and air are represented by $h_w$ and $h_a$, respectively. Similarly, $\theta_w$ and $\theta_a$ are the temperatures of water and air, respectively. Finally, $\theta_0$ denotes the initial temperature of the fibre. Radial temperature $\theta(r, t)$ is obtained by solving the above set of one-dimensional partial differential equations (see Supplementary Note 2 for more details).

Thereafter, we constructed a simple axial temperature model dominated by convection due to water transport since the flow rate of the water is much higher than the axial heat conduction rate of the fibre. The basis for this assumption is as follows. First, the high aspect ratio of the hollow fibre makes the heating/cooling process much faster in the radial direction than in the axial direction of the hollow-fibre actuator when considering only heat conduction. Second, polymers are poor heat conductors. When analysing the heat transfer mechanisms in the axial direction of the actuator, convection due to the fast water transport plays a major role compared to the slow heat conduction inside the fibre,

which is evident from Supplementary Fig. 3a: the actuation response rate is very sensitive to the water flow rate.

Water temperature $\theta_w(x, t)$ depends on a few factors, for instance, the initial water temperature, water flow rate, and heat loss due to transfer to the hollow-fibre actuator and eventually to the ambient air. Since the heat loss is relatively low when the flow rate is high, we can simply approximate the water temperature as

$$\theta_w(x, t) = \theta_{w0}H(\nu t - x), \tag{2}$$

where $\theta_{w0}$ denotes the initial water temperature, $\nu$ is the water flow rate, and $H(\cdot)$ is the Heaviside step function. By combining Eqs. (1) and (2), we determined the temperature of the hollow-fibre actuator as a function of the axial and radial positions and time, i.e., $\theta = \theta(x, r, t)$.

The torsional actuation mechanism of twisted polymer fibres has also been studied[39]. Upon heating, the polymer chains expand in the radial direction and contract in the axial direction, which can be accommodated by changes in the length, diameter, and twist of the hollow fibre. The change in length is considered to be small, and the following equation for untwisting can be obtained[58]:

$$\Delta T = \left(\frac{\Delta\lambda}{\lambda}\frac{1}{\cos^2\alpha_f} - \frac{\Delta d}{d}\right)T, \tag{3}$$

where $T$ denotes the twist density, $\Delta T$ denotes the untwist during torsional rotation, $d$ and $\Delta d$ are the diameter of the hollow fibre and its change, respectively, and $\lambda$ and $\Delta\lambda$ are the length and length change of the helically oriented polymer chain, respectively. Furthermore, this equation can be rewritten as

follows:

$$\Delta T = \left( \alpha_\lambda \frac{1}{\cos^2 \alpha_f} - \alpha_d \right) \Delta\theta \cdot T \tag{4}$$

where $\alpha_\lambda = -(5.3 \pm 0.4) \times 10^{-4} \mathrm{K}^{-1}$ and $\alpha_d = (5.2 \pm 0.7) \times 10^{-4} \mathrm{K}^{-1}$ are the respective axial and radial thermal expansion coefficients of PEHFs obtained from thermomechanical analysis (TMA). In this work, the average exterior hollow-fibre temperature change $\Delta\theta = \theta_w(x, t)$ is employed as the actuation temperature.

The tensile actuation mechanism of a coiled polymer fibre is well documented in Ref. [58]. and is described as follows:

$$\frac{\Delta L}{L} = \frac{l^2}{NL} \Delta T \tag{5}$$

where $L$ and $\Delta L$ denote the coil length and change in the coil length, respectively; $l$ is the length of the twisted hollow fibre; and $N$ is the number of coil turns.

Using these equations, the torsional and tensile actuation of hollow-fibre actuators were theoretically obtained. The temperature as a function of radial and axial positions and time can be solved using Eqs. (1) and (2). Then, the torsional actuation is obtained from Eq. (4). Finally, the tensile actuation is given by Eq. (5). The temperature solutions to Eqs. (1) and (2) are numerically solved in this work.

These theoretical results are consistent with the experimental results (Figs. 2e, f, 3a, c, and Supplementary Fig. 6c). The percent length change ($\Delta L/L$) of the tensile actuator is linearly related to the twist change ($\Delta T$) of the twisted hollow fibre (Eq. 5), which is monotonically correlated with the temperature change ($\Delta\theta$) of the hollow-fibre actuator (Eq. 4). Forced convection of axial water flow contributes to the actuator temperature, which monotonically correlates with the initial water temperature and water flow rate (Eq. 2). Therefore, we observe that the actuation stroke and response time are monotonically correlated with the water temperature and water flow rate (Figs. 2e, 3c, d and Supplementary Figs. 3b and 7a).

The response time can be related to the water temperature by examining the transient heat conduction equation (Eq. 1a in this context). A higher water temperature will provide a sharper temperature gradient. According to Eq. (1a), this will lead to a higher rate of change for the temperature.

### Hollow-fibre actuators for microfluidic sensing and manipulation

In high-throughput biosensing or medicine synthesis, a liquid containing specific chemicals is transported to certain vessels for chemical reactions[78,79]. Because temperature is an important factor influencing reactions, ensuring that the liquid with the required temperature flows into the desired vessel is important for precise control of chemical reactions[80,81]. However, during microfluid transport, the temperature may vary over time. Therefore, sensing the liquid temperature and sorting the liquid into the desired vessels are highly desirable for microfluidic manipulation. In the conventional design, the sorting devices contain different modules, such as sensors (for temperature measurement) and motors (for moving the tubes for transporting liquid) (Supplementary Fig. 10a), which require complex design and lead to a bulky size.

In this work, torsional and tensile hollow-fibre actuators were employed for microfluidic manipulation, which can precisely rotate and translate the transported liquid according to the liquid temperature (Supplementary Movies 2 to 4). The torsional hollow-fibre actuator can rotate to different angles when flowing liquid at different temperatures (Fig. 5a). Here, a 20-cm-long torsional PEHF$_{580-990}$ actuator with an inserted twist density of 400 turns m$^{-1}$ was provided for torsional microfluidic manipulation. A 96-well plate was placed below the PEHF$_{580-990}$ actuator to collect the liquid, which was transported at a flow rate of 100 mg s$^{-1}$. For a 0.5 K fluid temperature change (e.g., from 30 to 30.5 °C), the actuator rotated by 20° (Fig. 5b). The temperature resolution and normalised torsional rotation per Kelvin of this actuator are better than those of other thermally driven twisted actuators in previous studies[39,63,82–87] (Fig. 5c). Further changes in temperature resulted in a linear increase in the rotation angle with fluid temperature (Fig. 5d). Consequently, fluids with different temperatures can be sorted into the desired wells. Similarly, the homochiral hollow-fibre actuator can contract to different lengths when transporting liquids at different temperatures (Fig. 1b and Supplementary Fig. 10b), and a contractile actuator can transport and sort a liquid at different temperatures by contracting to different lengths (Supplementary Figs. 10c, d and Supplementary Movie 4).

We also demonstrated employment of the hollow-fibre actuator as a clamp to capture an object. A homochiral PEHF$_{580-990}$ actuator was wrapped around a mandrel and thermally annealed at 108 °C for 1 h to set the shape, and it was used as a soft clamp to capture a 2-g load. By flowing 80 °C water at a flow rate of 1.72 g s$^{-1}$, the actuator shrank and tightly trapped the load during continuous water flow, which was then lifted by hand. Stopping the supply of the water flow resulted in a temperature decrease of the actuator and release of the load (Supplementary Movie 5). This provides the possibility for hollow-fibre actuators to be employed in soft robotics and industrial manufacturing as artificial muscles with fast response speed and large stroke.

### Discussions

Compared to actuators driven by electroheating or air heating, fluid-flow-driven hollow-fibre actuators require setups for pumping, heating, and storing the liquid. These requirements can limit the application, and addressing these requirements would facilitate application of hollow-fibre actuators in a wide range of scenarios and provide additional opportunities for the design of fluid-driven smart materials and devices.

We investigated the dependence of the stability of the fluid-driven hollow-fibre actuators on the environmental conditions (temperature, moisture, and wind) and type of fluid (Supplementary Fig. 2). The results indicate that the relative change in rotation angle is less than 5% for the investigated experimental conditions, indicating that stable actuation performance was obtained. This agrees with the theoretical understanding, in which the heat transfer coefficient for air convection is 1-2 orders of magnitude smaller than that for convective heat transfer of liquid. Therefore, such liquid-flow-driven actuation of a hollow-fibre actuator showed good tolerance against environmental changes. To decrease the disturbance of actuation caused by environmental fluctuations, use of a high flow speed, a high actuation temperature, or a large-size hollow-fibre actuator is suggested. We also examined the actuation dependence on the type of applied fluid, fluid viscosity, and fluid density, and there is a negligible change in the actuation performance when actuated by the different investigated types of fluid (Supplementary Figs. 2d and 4). Decreasing the cooling temperature resulted in decreased hysteresis during heating/cooling cycles (Supplementary Fig. 5).

We summarise the response time of the hollow-fibre actuators investigated in this work in two tables. The response time is in the range of 0.88 to 3 s (Supplementary Tables 1 and 2), which are among the best values reported for thermally driven actuators

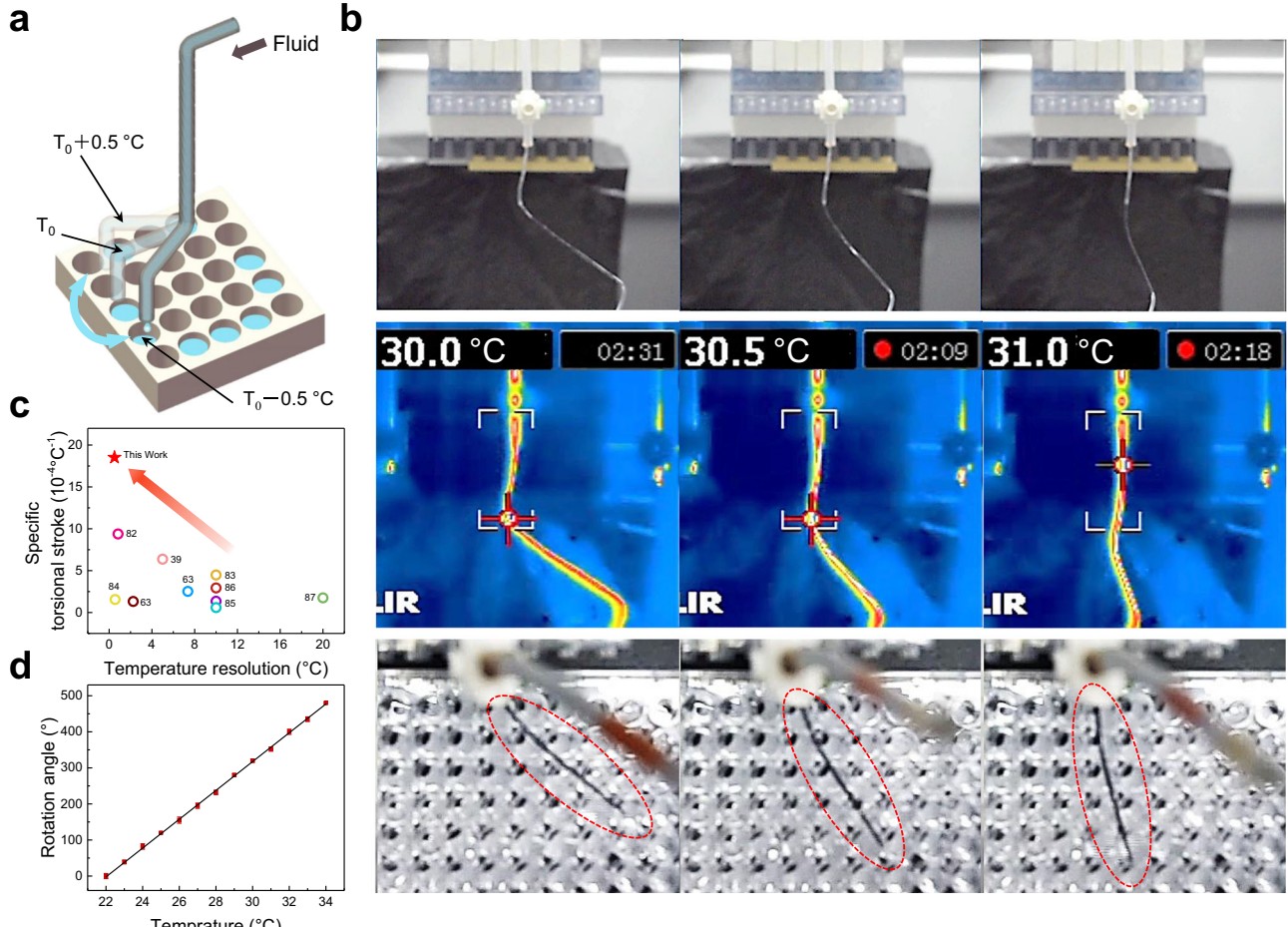

**Fig. 5 Microfluidic sensing and manipulation by PEHF$_{580-990}$ actuators. a** Schematic of the microfluidic sensing and manipulation device. **b** Optical images of PEHF$_{580-990}$ actuators for sensing the liquid temperature and sorting the liquid into the desired vessels. **c** Comparison of the temperature resolution and specific torsional stroke of PEHF$_{580-990}$ with those in previous studies. The specific torsional stroke is defined as $\Delta T/(T \cdot \Delta \theta)$, where $\Delta T$ denotes the thermally driven torsional rotation, $T$ denotes the twist density, and $\Delta \theta$ is the temperature change. The temperature resolution is the minimum temperature change used to trigger torsional actuation. **d** Torsional angle as a function of water temperature.

(Fig. 3f). The response time can be decreased by enhancing the forced convection between the fluid and hollow fibre and the heat transfer in the hollow fibre. These enhancements can be realised by increasing the fluid flow rate and fluid temperature, decreasing the wall thickness, and decreasing the fibre length of the hollow-fibre actuator.

From Eq. (5), we conclude that the resolution of the tensile actuation of the coiled hollow-fibre actuator ($\Delta L/L$) is proportional to $l^2 \Delta T/NL$, and from Eq. (4), we can obtain $\Delta T$; therefore, the $\Delta L/L$ for a temperature change of 1 K can be obtained from Eqs. (4) and (5) (where $l$ is the fibre length, $L$ and $\Delta L$ are the coil length and its change, respectively, $\Delta T$ is the change in twist density, and $N$ is the coil number). Supplementary Table 3 shows the relevant parameters of the homochiral hollow-fibre muscles and the resolution for a temperature change of 1 K. The length change of the muscles is up to −3% in this work.

We tried to obtain the optimised geometry of the hollow fibre (Supplementary Note 3). From the equation (Supplementary Eq. 3.4), the material should be distributed as far as possible from the centre of the cross-section, which justifies the use of hollow fibre over solid cross-sections. Note that the above theoretically derived optimum geometry is subject to the constraint that the hollow fibre will not buckle under the applied torque. Thus, in

practice, a hollow fibre with finite thickness should be used for the proposed actuation application.

In conclusion, a compact microfluidic manipulation system that can sense the liquid temperature and sort the liquid was developed by employing spiral hollow-fibre actuators. The fast sensing and manipulation originated from the indispensable combination of stress generation in the spiral hollow fibre sheath and efficient heat transfer during liquid flow. Additionally, this strategy is readily applicable to existing microfluidic tubing materials. In response to fluid flow, the resulting actuator could function with a high actuation stroke (87.5%), a fast response speed (0.88 s), and high temperature sensitivity (20° rotation for a temperature change of 0.5 K). The work capacity and power density were 1.5 times and 90 times those of air-driven solid fibres, respectively. This microfluidic manipulation design could be a new platform for high-throughput biosensing and medicine synthesis. The fast response and high power density of the fluid-driven actuator ensure its application in exoskeletons and artificial muscles, for instance, for developing powerful soft robots and automatic production lines. Moreover, the unique design provides new opportunities for developing equipment for biomarker sensing and drug control release, 3D printers, morphing aircraft, intelligent buildings, and other optical or magnetic applications.

## Methods

### Materials and methods

*Preparation of the hollow fibre actuators.* $PEHF_{280-640}$, $PEHF_{580-990}$, and $PEHF_{380-1090}$ and solid PE fibre (with a diameter of 1 mm) were obtained from Yongfa Plastic Products Co. Ltd). Nylon 6 hollow fibre (with inner and outer diameters of 0.1 mm and 0.4 mm) and nylon 6 fibre (with a diameter of 0.4 mm) are purchased from Sensa Fishing Import and Export Co., Ltd. All the hollow fibres were used as received.

Different types of fluids were employed to drive the actuation of the hollow fibre actuator, including ethanol (EtOH), iso-propanol (IPA), glycerol, acetone, dimethyl sulfoxide (DMSO), N, N-dimethylformamide (DMF), dichloromethane (DCM), tetrahydrofuran (THF), toluene, ethyl acetate (EA), and petroleum ether (PE), which were obtained from Shanghai Aladdin Biochemical Technology Co., Ltd. NaCl (Shanghai Macklin Biochemical Co., Ltd) was employed to prepare the fluid with different densities.

Before twist insertion, the hollow fibre was inserted with a copper wire (the diameter was slightly smaller than that of the inner diameter of the hollow fibre). The top end of the copper-wire-containing hollow fibre was connected to a 42-step servomotor (Huatian Technology Co., Ltd.), and its bottom end was isobarically loaded with 0.53 MPa weight. The load was torsionally tethered during twist insertion to avoid twist release[50]. During twist insertion, the bottom end of the PEHF was tethered and twist was inserted from the top side. The bottom end of the copper wire was not tethered and allowed to rotate during twist insertion, which enabled it to be easily removed after sample preparation. After twist insertion, the copper-wire-containing PEHF was both ends tethered and thermally annealed in an oven at a 108 °C for 1 h to thermally set the shape. The torsional hollow fibre actuator was obtained after cooling to room temperature and removing the copper wire. The coiled PEHF actuator was prepared by wrapping the twisted, copper-wire-containing hollow fibre on a mandrel, followed by annealing at 108 °C for 1 h. The solid PE fibre and the nylon 6 hollow fibre were twisted in the same way, and the tensile coiled nylon 6 actuators were prepared by twisting the nylon 6 fibre until self-coiling under a 23.4 MPa stress and annealing at 180 °C for 2 h.

*Actuation of the hollow fibre actuators.* Water-driven actuation was carried out by pumping water into the twisted or coiled hollow fibres using a water pump (Mini K- DCS10, Kamoer Fluid Tech Co., Ltd.) Actuation driven by hot air was carried out in a temperature-controlled oven. A thermal camera (FLIR T440) was used to monitor the surface temperature of the hollow fibre during actuation. The surface of the hollow fibre showed uniform temperature during actuation by water flow.

In this study different parameters were studied. While if not specified, the water temperature was 95 °C, and the water flow rate was 0.5 g s$^{-1}$; the torsional hollow fibre actuator was a $PEHF_{580-990}$ actuator with twist density of 400 turns mm$^{-1}$; the homochiral hollow fibre actuator was a $PEHF_{580-990}$ actuator with spring index of 4.0, coil pitch of 9 mm, and the twist density was 200 turns m$^{-1}$; the heterochiral hollow fibre actuator was a $PEHF_{580-990}$ actuator with spring index of 4.0, no coil pitch, and the twist density was 200 turns m$^{-1}$. The hollow-fibre actuators were trained for 3 cycles to obtain repetitive actuation performances.

## Data availability

The authors declare that all data supporting the findings of this study are available within the main text and Supplementary Information. Source data are provided with this paper.

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

## Acknowledgements

This work was supported by the National Key Research and Development Program of China (grants 2019YFE0119600), the National Natural Science Foundation of China (grants 51973093, 51773094, and U1533122), Frontiers Science Center for New Organic Matter, Nankai University (Grant Number 63181206), the National Special Support Plan for High-Level Talents People, the Science Foundation for Distinguished Young Scholars of Tianjin (grant no. 18JCJQJC46600), the Xingliao Talent Plan (XLYC1802042), the Fundamental Research Funds for the Central Universities (grant 63171219), the State Key Laboratory for Modification of Chemical Fibers and Polymer Materials, Donghua University (grant LK1704), the grant from the US National Science Foundation (Award CMMI-1727960), and Eugene McDermott Graduate Fellowship at The University of Texas at Dallas.

## Author contributions

Z.F.L. and S.L. were responsible for the experimental concept and design. S.L., G.Z., L.S, W.C., X.Z., X.H., B.A. and M.Z. carried out the most experiments, characterisation and data analyses. R.Z. and D.Q. contributed to theoretical simulation and calculation. Z.F.L. was responsible for project administration, conceptualisation, supervision, formal analysis, funding acquisition, validation, writing original draft, review and editing. All authors wrote the paper. All authors provided comments and agreed with the final form of the manuscript.

## Competing interests

The authors declare no competing interests.
