## [Peer Review File · Nature Communications]

REVIEWER COMMENTS

Reviewer #1 (Remarks to the Author):

This paper presents the development of fast torsional and tensile actuators using hollow fibers employing spiral nonlinear stress sensing the fluid temperature and sort the fluid into the desired vessels. Various applications were demonstrated to provide a new design strategy for intelligent microfluidics and provides inspiration for soft robotics and smart devices for bio-, opto-, or magneto applications.

The concept of intelligent microfluidic system looks pretty interesting and the combination of microfluidic and the temperature operated actuator sounds very clever. This paper can be recommended for publication with the following comments,

1) Inserting a twist into hollow fibers may generate a spiral hollow fiber actuator and, a flowing liquid that can cause hollow-fiber volume expansion resulted in position control of the twisted hollow fiber. I guess this is the first demonstration of fluid operated hollow fiber actuator. The analysis and application demonstration look very promising. In addition to the novelty, it looks like there is some limitation in this kind of actuator that is operated by the temperature of the fluid. Cause the fluid should flow inside the hollow actuator, that flow should move to some place (it may come out of the microfluidic system or go the some external fluidic reservoirs). If the authors can further discuss the limitation of this system (or comparison with other types of thermally driven actuators), it will provide a useful perspective to the readers.

2) As the authors discussed, a thinner fiber would rotate more than a thicker fiber for the same change in the bias angle. However, besides the mechanical property of the fiber, the major mechanism of this thermal actuator is the heat transfer from the liquid transportation in the hollow fiber. Basically, this is a typical heat transfer problem in a pipe heat exchanger. For the cylindrical pipe, the heat transfer can be calculated by a simple thermal circuit analysis (refer to the chapter 3.3 in the Principals of Heat and Mass Transfer by Incropera). The heat transfer will be dependent on the pipe (hollow fiber in this study) diameter and thickness and the optimum thickness (critical radius) may be obtained. Even though the current fiber geometry may be far different from the calculated optimum value, it would be worth to provide the analysis approach for the optimum hollow fiber geometry.

3) The potential problem of the thermally operated actuator is the actuation performance may be influenced by the environment condition (temperature, wind, weather etc.). I wonder how stable the proposed system to the environment and how this can be improved. The related discussed will be very useful to the practical usage.

4) The authors discussed that the actuation stroke and response time were monotonically correlated with the water temperature (Supplementary Figs. 3b and c). Any discussion on this observation on the monotonic correlation with water temperature will be needed.

5) The fast response and large actuation stroke of the coiled hollow-fiber actuators when the transported fluid changes its temperature would allow precise sensing and microfluidic manipulation. More discussion on this?

- 6) The application demonstration was well chosen and interesting. I guess most of the application is related to the dispensing of the liquid. Is there any other example of other applications?
- 7) There are several different types of thermally driven dry state actuators such as electrothermal actuator (Adv. Funct. Mater., 28, 1801847, 2018.), photothermal actuator (Adv. Funct. Mat. 29, 1808995, 2019), shape memory actuator etc. These need to be discussed in the introduction part.
- 8) As the hollow-fiber actuator was driven by flowing hot liquid, theoretically correlating the actuation behavior with heat transfer is highly desirable to understand the actuation mechanism for designing hollow-fiber actuators for precise microfluidic manipulation. Therefore, the authors performed the thermal-mechanical modeling of the torsional and tensile hollow-fiber actuators by quantifying the heat transfer between the flowing liquid and hollow fiber. I wonder whether the temperature dependent material properties were considered in this analysis or not.
- 9) Was there any actuation dependence on the type of the applied fluid?
- 10) Usually, thermally driven actuator has relatively slower response. What is the general response time of the actuator and how this can be enhanced?
- 11) Some typos were observed in this manuscript (for example, “microfluidic contro^{23,24}” in introduction in page 2) and the general language usage including typos and grammatical errors should be checked.

Reviewer #2 (Remarks to the Author):

This paper demonstrates an interesting way of using twisted or coiled polymer fibre artificial muscles in microfluidic applications. Commercial hollow fibres are used to torsionally deform the sheath supported by solid metal wire used in the core, inspired by the popular way of making twisted fibre actuators. Although adequate experimental results are presented, the fundamental actuation mechanism is incorrectly picked in some statements (are the actuators really fluid driven?). Highly repetitive actuation cycles are obtained without training the newly made actuators, which I find surprising for the demonstrated untethered system. While the fundamental actuation mechanism was kept aligned to the well-established single helix model of solid twisted/coiled actuators, the demonstrated hollow fibres might show deviated estimation as the helices are not reached the fibre axis. Please refer to the below specific comments to work with:

1. Line 59-60: How flowing liquid can cause volume expansion of the hollow fibre unless having a moisture responsive filler? According to the principle of McKibben muscle, a tubular shape can be expanded while pressurized fluid is introduced. Are they hydrothermal driven, thermal stimulus causing the shape deformation? Is fluid having no driving effect on the actuation?

2. Line 60-61: How a solid fibre can transport and manipulate microfluidics?
3. Line 62-64: Add references where air-driven actuators have been used to transport liquids.
4. Line 64-66: Although I do agree that fibre surface have higher bias angle than the centre, but a solid fibre can only have centre with physical bias angle. How a hollow fibre can have bias angle, air cannot be physically twisted.
5. Line 70: Twist configured thermoplastic fibres can be heat set above their T_g , to prevent twist loss. This is a well-established method, as used in this paper, so nothing new is to be addressed.
6. Line 100-102: There should be some extent of twist loss, when the fibre was released from tethered condition. This might be attributed to the stress relaxation phenomenon of deformed polymer shapes. It is necessary to count those twists loss (even they are small) and consider the number of remaining twists for calculating twist density (as demonstrated in SI Figure 5).
7. Line 103-105: Just inserting twists into a polymer fibre is not the only criteria that give them actuation capability. The fibres should be of highly oriented semicrystalline form. How do the authors know the used commercial hollow fibres have such structure. Experimental evidence is needed.
8. Line 149-158: These are well-known information from many previously published articles. Please remove from the main manuscript, this information better suited in SI.
9. Line 163-164: How is the response time (1.2 sec) considered? The flow of hot and cold fluid from one end to the other end will certainly have time-dependent temperature gradients. It seems like fluid pass will take more than the response time, please explain. Also, explain the effect of the mass of the weight flown through. Is there any hysteresis of actuation present between cooling and heating cycles? If present, how about using cold (e.g. ice water) fluid to reduce the hysteresis.
10. Line 169-170: Coiled polymer fibre actuators usually need a few training cycles before having fully reversible actuation cycles (no or insignificant decay). I am not quite convinced that these actuators show no decay from the very first cycle. Please explain the method of how the non-decayed cycles were received from the beginning. Was there any special procedure followed to overcome the creeping effect? The same question also applied for Line 210-213, where effect of different stress on actuation performance is demonstrated. Repetitive (better to name 'reversible') actuation is certainly possible, but not before training cycles or any novel method applied.

11. Line 243-245: Why is heat conduction rate along the hollow fibre axis is important? Most of the heat from the hollow core to the polymer sheath might transfer through convection via the fluid. Please explain the significance.

12. Line 248-250: Please explain the basis of the assumption taken? Is there any relevant reference article?

13. Line 261-266: This model used solid twisted fibres. It is possible that the hollow core twisted fibres follow slightly different torsional actuation mechanisms. If not, please explain the reason.

14. Line 269-270: Please explain the method of measuring the thermal expansion coefficient of the PE used. Differently drawn semicrystalline polymers will have different coefficients, so simply using reference values will not be feasible for highly sophisticated microfluid manipulation devices. Please include relevant results found from all samples used.

15. Line 278-279: Is it possible to develop a consolidated equation? Or a step-by-step method? I really don't understand how to correlate the heat transfer equation with the other three reference equations, for theoretically predicting the actuation results.

16. Line 284-290: Provide appropriate references for these statements.

17. Line 305-306: What does it mean by 'linear increase in the rotation angle'? Fig. 5d should be modified based on the explanation provided, why the angle goes from negative to positive in the current figure is not understandable.

18. Line 387-390: How the 'isobaric stresses' can be explained, when the coiled fibre diameter will change during contraction and expansion? Was there the dynamic change of pressure (stress) considered? Or the helical hollow fibre diameter was used, which does not make sense for such a vertically suspended system.

Overall, proofread the paper carefully, current version consists of many statements that are incorrect, rather empty, or ambiguous.

Reviewer #3 (Remarks to the Author):

This paper presents fast torsional and tensile actuators made of hollow fibers, which can detect the fluid temperature and sort the fluid into the desired vessels. It is interesting and well written. Experiments are well planned and conducted. Analysis and conclusions are sound. It shows great application prospects.

Some minor problems should be addressed by the authors as follows:

1. "The fluid-driven actuation was 27 times as that driven by air flow; the work capacity and power density were 1.5 times and 90 times, respectively". This reviewer can see where the data supporting the statement from Fig. 3F and fig. 4F. Can the authors move the supplementary information in the main text as this is an important statement.
2. "To facilitate counting of the rotation angle, a paddle (with a mass of 1/50 of the hollow fiber) was taped at the outlet of the PEHF580-990 (Fig. 2b)". Please indicate the paddle by a clear method as this reviewer cannot find it in Fig. 2b.
3. In Supporting information video1: On the upper left part, leakage occurs and water jet out of the actuator (not the droplets). It seems that the actuator has a pole. Please explain this phenomenon.
4. The fluid in this article is mainly water, but there is no discussion on other fluids. There are many kinds of fluids, some of which are more viscous and not suitable for such a small-diameter pipe. Please explain the range of practical fluids at the end of the summary, or summarize the prospect.
5. Supplementary Figure 1. (a). It is better to delete the data at the end of this curve (the vertical part). Because the data of this part were measured when the fiber was cracked, thus the slope is nearly ∞ .
6. "In recent years, stimuli-responsive materials have been incorporated into micrometer channels to realize microfluidic contro" (missing letter 'l', please check the whole paper that there is no cacography!)

Response to the reviewers:

Response to the reviewer #1:

This paper presents the development of fast torsional and tensile actuators using hollow fibers employing spiral nonlinear stress sensing the fluid temperature and sort the fluid into the desired vessels. Various applications were demonstrated to provide a new design strategy for intelligent microfluidics and provides inspiration for soft robotics and smart devices for bio-, opto-, or magneto applications.

The concept of intelligent microfluidic system looks pretty interesting and the combination of microfluidic and the temperature operated actuator sounds very clever. This paper can be recommended for publication with the following comments,

Response: We thank the reviewer for this insightful comment, and we have made a substantial revision to improve this manuscript based on the reviewers' comments.

1) Inserting a twist into hollow fibers may generate a spiral hollow fiber actuator and, a flowing liquid that can cause hollow-fiber volume expansion resulted in position control of the twisted hollow fiber. I guess this is the first demonstration of fluid operated hollow fiber actuator. The analysis and application demonstration look very promising. In addition to the novelty, it looks like there is some limitation in this kind of actuator that is operated by the temperature of the fluid. Cause the fluid should flow inside the hollow actuator, that flow should move to some place (it may come out of the microfluidic system or go the some external fluidic reservoirs). If the authors can further discuss the limitation of this system (or comparison with other types of thermally driven actuators), it will provide a useful perspective to the readers.

Response: We thank the reviewer for this suggestion. We discussed the limitations of such a fluid-flow-driven hollow fiber actuator, as follows. (Page 13, Line 381-385)

Compared to actuators driven by electroheating or air heating, fluid-flow-driven hollow-fibre actuators require setups for pumping, heating, and storing the liquid. These requirements can limit the application, and addressing these requirements

would facilitate application of hollow-fibre actuators in a wide range of scenarios and provide additional opportunities for the design of fluid-driven smart materials and devices.

2) As the authors discussed, a thinner fiber would rotate more than a thicker fiber for the same change in the bias angle. However, besides the mechanical property of the fiber, the major mechanism of this thermal actuator is the heat transfer from the liquid transportation in the hollow fiber. Basically, this is a typical heat transfer problem in a pipe heat exchanger. For the cylindrical pipe, the heat transfer can be calculated by a simple thermal circuit analysis (refer to the chapter 3.3 in the Principals of Heat and Mass Transfer by Incropera). The heat transfer will be dependent on the pipe (hollow fiber in this study) diameter and thickness and the optimum thickness (critical radius) may be obtained. Even though the current fiber geometry may be far different from the calculated optimum value, it would be worth to provide the analysis approach for the optimum hollow fiber geometry.

Response: We thank the reviewer for this comment and providing the reference book on thermal analysis. The main equation from chapter 3.3 of Incropera's book provides the steady-state solution for the case of hollow cylinder with convective surface conditions. While the axisymmetric configuration considered in the book can be directly extended, this analytical solution does not give the time-dependence of the temperature field that is important for estimating the response time of our actuator. Since we have a moving heat source with time-varying temperatures, the heat transfer in our system is transient in nature and is difficult to be solved analytically.

To provide an estimation of the optimum configuration of the hollow tube, we assume a steady heat solution and constant temperature due to the small thickness of the tube. Based on the linear elastic solution for a thin-walled tube under pure torsion [Book: R. G. Budynas, J. K. Nisbett, Shigley's Mechanical Engineering Design (McGraw-Hill, New York, 2008).], the generated torque M is related to the twist through

$$M = 2\pi GJT, \quad (3.1)$$

in which G is the shear modulus, J is the polar second moment of area. For the hollow tube section $J = \frac{\pi}{2}(r_o^4 - r_i^4)$ where r_i and r_o are the inner and outer radius of the hollow tube, respectively.

Applying Eq. (3.1) to the tube configurations before and after the torsional actuation provides the torque release ΔM :

$$\Delta M = \pi G A_0 (r_i^2 + r_o^2) [(I + \alpha_d \Delta \theta)^4 (T + \Delta T) - T], \quad (3.2)$$

in which A_0 is the cross-section area of the hollow tube.

We now introduce a simplified version of torsional actuation based on the reference (*J. Polym. Sci. B*, 54, 1278, 2016):

$$\Delta T = \left(\frac{d_0}{d} - 1\right) T. \quad (3.3)$$

Substituting Eq. (3.3) into (3.2) gives

$$\Delta M = \pi G A_0 (r_i^2 + r_o^2) [(I + \alpha_d \Delta \theta)^3 - 1] T. \quad (3.4)$$

Based on Eq. (3.4), we conclude that the optimum geometry for maximizing the torque output is to maximize $(r_i^2 + r_o^2)$, assuming that the cross-section area A_0 (or equivalently the material consumption) is constant. This means that the material should be distributed as far away as possible from the center of the cross-section, which justifies the use of a hollow tube over solid cross-sections. It should be noted that the above theoretically derived optimum geometry is subjected to the constraint that the tube will not buckle under the applied torque. Thus, in practice a tube with finite thickness should be used for the proposed actuation application.

We added these discussions in the Supplementary Note 3, and in the main text (Page 15, Line 417-423).

3) The potential problem of the thermally operated actuator is the actuation performance may be influenced by the environment condition (temperature, wind, weather etc.). I wonder how stable the proposed system to the environment and how this can be improved. The related discussed will be very useful to the practical usage.

Response: We added actuation experiments of the hollow-fiber actuators in different environmental conditions (temperature, moisture, and wind), we also added actuation experiments by different types of fluid (Supplementary Fig. 2). The results indicate that the relative change in rotation angle is less than 5% for the investigated experimental conditions, indicating that stable actuation performance was obtained. This agrees with the theoretical understanding, in which the heat transfer coefficient for air convection is 1-2 orders of magnitude smaller than that for convective heat transfer of liquid. Therefore, such liquid-flow-driven actuation of a hollow-fibre actuator showed good tolerance against environmental changes. To decrease the disturbance of actuation caused by environmental fluctuations, use of a high flow speed, a high actuation temperature, or a large-size hollow-fibre actuator is suggested. We added this discussion in the context (Page 14, Line 386-396).

Supplementary Figure 2. The torsional actuation of the PEHF₅₈₀₋₉₉₀ actuator by flowing hot water at 1.72 g s⁻¹ at different actuation conditions: (a) at different environment temperatures, (b) at different environmental relative humidity, and (c) at different wind speed. (d) The torsional actuation of the PEHF₅₈₀₋₉₉₀ actuator by

flowing different types of 40 °C liquid at a flow rate of 1.72 g s⁻¹. The δ and $\Delta\delta$ are the rotation angle during actuation and its change. If not specified, the room temperature is 25 °C, the environmental relative humidity is 40%, and the water flow rate is 1.72 g s⁻¹. If not specified, error bars in this figure and in the following figures indicate standard deviations.

4) *The authors discussed that the actuation stroke and response time were monotonically correlated with the water temperature (Supplementary Figs. 3b and c). Any discussion on this observation on the monotonic correlation with water temperature will be needed.*

Response: Thank you for the suggestion. We added discussion about the monotonic correlation of the actuation stroke and response time with the water temperature (Page 12, Line 330-341), as follows.

The percent length change ($\Delta L/L$) of the tensile actuator is linearly related to the twist change (ΔT) of the twisted hollow fibre (Eq. 5), which is monotonically correlated with the temperature change ($\Delta\theta$) of the hollow-fibre actuator (Eq. 4). Forced convection of axial water flow contributes to the actuator temperature, which monotonically correlates with the initial water temperature and water flow rate (Eq. 2). Therefore, we observe that the actuation stroke and response time are monotonically correlated with the water temperature and water flow rate (Figs. 2e, 3c, 3d and Supplementary Figs. 3b, 7a).

The response time can be related to the water temperature by examining the transient heat conduction equation (Eq. 1a in this context). A higher water temperature will provide a sharper temperature gradient. According to Eq. (1a), this will lead to a higher rate of change for the temperature.

However, as mentioned in the answer to the second question, obtaining analytical expression on the dependence of water temperature on activation response time is difficult for this application.

$$\frac{\partial^2 \theta}{\partial r^2} + \frac{1}{r} \frac{\partial \theta}{\partial r} = \frac{1}{\alpha} \frac{\partial \theta}{\partial t} \quad \text{in } R_i \leq r \leq R_o \quad (1a)$$

$$\theta_w(x, t) = \theta_{w0} H(vt - x), \quad (2)$$

$$\Delta T = \left(\alpha_\lambda \frac{1}{\cos^2 \alpha_f} - \alpha_d \right) \Delta \theta \cdot T, \quad (4)$$

$$\frac{\Delta L}{L} = \frac{l^2}{NL} \Delta T, \quad (5)$$

5) *The fast response and large actuation stroke of the coiled hollow-fiber actuators when the transported fluid changes its temperature would allow precise sensing and microfluidic manipulation. More discussion on this?*

Response: This is a very good suggestion. We added discussion about “The fast response and large actuation stroke of the coiled hollow-fiber actuators when the transported fluid changes its temperature would allow precise sensing and microfluidic manipulation.” (Page 7, Line 202-209; Page 14-15, Line 413-419), as follows:

For example, the coiled hollow-fibre actuator would spontaneously change its length with the change in the flowing liquid temperature, thereby indicating the real-time temperature of the fluid; in addition, the large actuation stroke of the hollow fibre in response to the fluid temperature change would provide the possibility to sort the liquid with different temperatures into desired vessels. Such a fluid-driven actuator can be employed as a new type of artificial muscle with fast and large stroke in soft robots, manipulators in production lines, exoskeletons, and wearable power assist robots. (Page 7, Line 202-209)

From Eq. (5), we conclude that the resolution of the tensile actuation of the coiled hollow-fibre actuator ($\Delta L/L$) is proportional to $l^2 \Delta T/NL$, and from Eq. (4), we can obtain ΔT ; therefore, the $\Delta L/L$ for a temperature change of 1 K can be obtained from Eqs. (4) and (5) (where l is the fibre length, L and ΔL are the coil length and its change, respectively, ΔT is the change in twist density, and N is the coil number). Supplementary Table 3 shows the relevant parameters of the homochiral hollow-fibre muscles and the resolution for a temperature change of 1 K. (Page 14, Line 409-415)

$$\Delta T = \left(\alpha_\lambda \frac{1}{\cos^2 \alpha_f} - \alpha_d \right) \Delta \theta \cdot T, \quad (4)$$

$$\frac{\Delta L}{L} = \frac{l^2}{NL} \Delta T. \quad (5)$$

Supplementary Table 3. Resolution and the coil parameters of the homochiral PEHF hollow fibre actuators in this work

Fibre Length (l , mm)	Coil Length (L , mm)	Number of Turns (N)	Twist density (T , turns mm^{-1})	Resolution ($\Delta L/L$)
200	105	11	0.30	-3.0%
200	105	11	0.25	-2.0%
200	105	11	0.20	-1.3%
200	105	11	0.15	-0.78%
200	105	11	0.10	-0.43%
200	115	12	0.30	-2.5%
200	115	12	0.25	-1.7%
200	115	12	0.20	-1.1%
200	115	12	0.15	-0.66%
200	115	12	0.10	-0.36%
200	130	15	0.25	-1.2%
200	130	15	0.20	-0.78%
200	130	15	0.10	-0.26%
200	145	18	0.20	-0.58%
200	145	18	0.10	-0.19%

6) *The application demonstration was well chosen and interesting. I guess most of the application is related to the dispensing of the liquid. Is there any other example of other applications?*

Response:

We also demonstrated employment of the hollow-fibre actuator as a clamp to capture an object. A homochiral PEHF₅₈₀₋₉₉₀ actuator was wrapped around a mandrel and thermally annealed at 108 °C for 1 h to set the shape, and it was used as a soft clamp to capture a 2-g load. By flowing 80 °C water at a flow rate of 1.72 g s⁻¹, the actuator shrank and tightly trapped the load during continuous water flow, which was then lifted by hand. Stopping the supply of the water flow resulted in a temperature decrease of the actuator and release of the load (Supplementary Movie 5). This provides the possibility for hollow-fibre actuators to be employed in soft robotics and

industrial manufacturing as artificial muscles with fast response speed and large stroke. We added the discussion in the context (Page 13, Line 371-379).

7) *There are several different types of thermally driven dry state actuators such as electrothermal actuator (Adv. Funct. Mater., 28, 1801847, 2018.), photothermal actuator (Adv. Funct. Mat. 29, 1808995, 2019), shape memory actuator etc. These need to be discussed in the introduction part.*

Response: We added discussion in the introduction part about these thermally driven actuators. (Page 2, Line 49, and 55-56).

8) *As the hollow-fiber actuator was driven by flowing hot liquid, theoretically correlating the actuation behavior with heat transfer is highly desirable to understand the actuation mechanism for designing hollow-fiber actuators for precise microfluidic manipulation. Therefore, the authors performed the thermal-mechanical modeling of the torsional and tensile hollow-fiber actuators by quantifying the heat transfer between the flowing liquid and hollow fiber. I wonder whether the temperature dependent material properties were considered in this analysis or not.*

Response: We thank the reviewer for this insightful question. For the thermal-mechanical modeling of the torsional and tensile hollow-fiber actuators, it is reported that the fiber thermal conductivity did not show a large change during the investigated temperature range (*J. Polym. Sci. Polym. Phys. Ed. 18*, 1187 (1980); *Polym. Test 32*, 987 (2013)), and we employed an average value of the measured thermal expansion coefficient for the model, which works well for the investigated temperature range.

9) *Was there any actuation dependence on the type of the applied fluid?*

Response: We added experiments to show the actuation dependence on the type of the applied fluid, fluid viscosity, and fluid density, and there is a negligible change in the actuation performance when actuated by the different investigated types of fluid (Supplementary Figs. 2d and 4). We added these discussions in the context, as follows

(Page 14, Line 397-399).

Supplementary Figure 2d. The torsional actuation of the PEHF₅₈₀₋₉₉₀ actuator by flowing different types of 40 °C liquid at 1.72 g s⁻¹. The room temperature is 25 °C, the environmental relative humidity is 40%. EtOH: ethanol; EA: ethyl acetate; DMSO: dimethyl sulfoxide; THF: tetrahydrofuran; DMF: N, N-dimethylformamide; DCM: dichloromethane; IPA: iso-propanol; PE: petroleum ether.

Supplementary Figure 4. (a) The initial coil length and actuation stroke of the

homochiral PEHF₅₈₀₋₉₉₀ actuator by flowing 25 °C glycerol/water solution with different viscosity. (b) Actuation stroke of the PEHF₅₈₀₋₉₉₀ actuator by flowing 80 °C glycerol/water solution with different viscosity. (c) The initial coil length and actuation stroke of the PEHF₅₈₀₋₉₉₀ actuator by flowing 25 °C aqueous NaCl solution with different densities. (d) Actuation stroke of the PEHF₅₈₀₋₉₉₀ actuator by flowing 80 °C aqueous NaCl solution with different densities. The spring index was 4.0, and the twist density was 200 turns m⁻¹.

10) Usually, thermally driven actuator has relatively slower response. What is the general response time of the actuator and how this can be enhanced?

Response: We added two tables to summarize the response time of the hollow-fiber actuators investigated in this work. The response time are in the range of 0.88 to 3 s (Supplementary Tables 1 and 2), which are among the best values reported for thermally driven actuators (Fig. 3f). The response time can be decreased by enhancing the forced convection between the fluid and hollow fibre and the heat transfer in the hollow fibre. These enhancements can be realized by increasing the fluid flow rate and fluid temperature, decreasing the wall thickness, and decreasing the fibre length of the hollow-fibre actuator. We added this discussion in the context. (Page 14, Line 402-408)

Supplementary Table 1. Response time and actuation parameters of the torsional PEHF actuators in this work

Materials	Twist density (turns m ⁻¹)	$\Delta\theta$ (°C)	Flowrate (g s ⁻¹)	Response time (s)
PEHF ₅₈₀₋₉₉₀	100	70	0.5	2.5
PEHF ₅₈₀₋₉₉₀	200	70	0.5	2.1
PEHF ₅₈₀₋₉₉₀	300	70	0.5	1.9
PEHF ₅₈₀₋₉₉₀	400	70	0.5	1.7
PEHF ₅₈₀₋₉₉₀	400	55	0.5	1.8
PEHF ₅₈₀₋₉₉₀	400	45	0.366	2.7

PEHF ₅₈₀₋₉₉₀	400	35	0.327	1.1
PEHF ₅₈₀₋₉₉₀	400	0.5	0.1	1.0

Supplementary Table 2. Response time and actuation parameters of the tensile PEHF actuators in this work

Materials	Twist density (turns m ⁻¹)	$\Delta\theta$ (°C)	Flowrate (g s ⁻¹)	Response time (s)
PEHF ₅₈₀₋₉₉₀	200	35	1.72	2.5
PEHF ₅₈₀₋₉₉₀	200	45	1.72	1.6
PEHF ₅₈₀₋₉₉₀	200	55	1.72	1.4
PEHF ₅₈₀₋₉₉₀	200	70	1.72	0.88
PEHF ₅₈₀₋₉₉₀	200	70	0.5	1.4
PEHF ₅₈₀₋₉₉₀	200	70	0.366	1.6
PEHF ₅₈₀₋₉₉₀	200	70	0.327	1.7
PEHF ₅₈₀₋₉₉₀	200	70	0.245	2.8
PEHF ₂₈₀₋₆₄₀	310	70	0.366	1.0
PEHF ₃₈₀₋₁₀₉₀	182	70	0.366	3.0

11) Some typos were observed in this manuscript (for example, “microfluidic contro23,24” in introduction in page 2) and the general language usage including typos and grammatical errors should be checked.

Response: We have corrected the typos and polished the English by language service.

This document certifies that the manuscript
Microfluidic Manipulation by Spiral Hollow-Fiber Actuators

prepared by the authors

Sitong Li, Rui Zhang, Guanghao Zhang, Luyizheng Shuai, Wang Chang, Xiaoyu Hu, Min
Zou, Xiang Zhou, Baigang An, Dong Qian, Zunfeng Liu*

was edited for proper English language, grammar, punctuation, spelling, and overall style
by one or more of the highly qualified native English speaking editors at SNAS.

This certificate was issued on **December 7, 2021** and may be verified
on the SNAS website using the verification code **E505-C78B-1FOC-CF9E-D36P**.

Neither the research content nor the authors' intentions were altered in any way during the editing process. Documents receiving this certification should be English-ready for publication; however, the author has the ability to accept or reject our suggestions and changes. To verify the final

SNAS edited version, please visit our verification page at secure.authorservices.springernature.com/certificate/verify.

If you have any questions or concerns about this edited document, please contact SNAS at support@as.springernature.com.

Response to the reviewer #2:

This paper demonstrates an interesting way of using twisted or coiled polymer fibre artificial muscles in microfluidic applications. Commercial hollow fibres are used to torsionally deform the sheath supported by solid metal wire used in the core, inspired by the popular way of making twisted fibre actuators. Although adequate experimental results are presented, the fundamental actuation mechanism is incorrectly picked in some statements (are the actuators really fluid driven?). Highly repetitive actuation cycles are obtained without training the newly made actuators, which I find surprising for the demonstrated untethered system. While the fundamental actuation mechanism was kept aligned to the well-established single helix model of solid twisted/coiled actuators, the demonstrated hollow fibres might show deviated estimation as the helices are not reached the fibre axis. Please refer to the below specific comments to work with:

Response: We thank the reviewer for these valuable comments and insightful suggestions, and we have carefully addressed these questions in the revised version of this article. Here we provide the point-to-point reply below.

1. Line 59-60:

(1) How flowing liquid can cause volume expansion of the hollow fibre unless having a moisture responsive filler? According to the principle of McKibben muscle, a tubular shape can be expanded while pressurized fluid is introduced. Are they hydrothermal driven, thermal stimulus causing the shape deformation? Is fluid having no driving effect on the actuation?

Response: We thank the reviewer for these insightful comments. We have re-written this paragraph to make it clear.

The actuation mechanism of the twisted hollow fibre is similar to that of a twisted solid fibre. When the fibre sheath is heated by the flowing hot liquid, the sheath material contracts in the axial direction and expands in the radial direction, causing torsional and tensile actuation for a twisted and a coiled hollow-fibre actuator, respectively. (Page 2, Line 58-61)

We hypothesize that the actuation mechanism in the proposed hollow-fibre actuator is different from that of a McKibben muscle, which is hydraulically driven. To test this hypothesis, we added an experiment to demonstrate the effect of water flow on the actuation performance. No actuation occurred when room-temperature fluid flowed through the twisted hollow fibre (Supplementary Figs. 3c and 4). Actuation occurred when a hot fluid flowed through the twisted hollow fibre, and the actuation stroke and actuation speed increased with increasing fluid temperature (Figs. 2e and 3c). These results show that the fluid-driven hollow-fibre actuator is hydrothermally driven, not hydraulically driven. (Page 8, Line 224-232)

In response to this question and the flowing questions 2 to 5, we have re-written this paragraph to make it clear as follows:

Twisted fibres can generate torsional and tensile actuation by volume expansion, and various fibre actuators have been developed⁴⁰⁻⁵⁶. In the present study, we envisage that inserting a twist into a hollow fibre may generate a spiral hollow-fibre actuator. The actuation mechanism of the twisted hollow fibre is similar to that of a twisted solid fibre. When the fibre sheath is heated by the flowing hot liquid, the sheath material contracts in the axial direction and expands in the radial direction, causing torsional and tensile actuation for a twisted and a coiled hollow-fibre actuator, respectively. Consequently, the actuation of the hollow fibre in response to a liquid with different temperatures would result in microfluidic position control. However, to date, there have been no reports on the incorporation of twist insertion into hollow fibres for actuators. This difficulty may be caused by the buckling and collapse of the hollow fibre during twist insertion, which blocks liquid flow.

Liquid transport in the hollow fibre would result in efficient heat transfer at the fluid-hollow fibre interface, and consequently, a faster response would be expected. Furthermore, the outer surface of a fibre always exhibits a higher bias angle than the inner fibre surface⁵⁷, and consequently, increased actuation stress and work output are expected for a hollow fibre with the same mass but a larger outer radius (Supplementary Note 3). (Pages 2-3, Line 55-72)

2. Line 60-61: *How a solid fibre can transport and manipulate microfluidics?*

Response: We thank the reviewer to point out this point, and we have removed this sentence from this paragraph.

3. Line 62-64: *Add references where air-driven actuators have been used to transport liquids.*

Response: We thank the reviewer to point out this point. We have re-write this sentence to clearly express our meaning (Page 3, Line 67-68), as follows:

Liquid transport in the hollow fibre would result in efficient heat transfer at the fluid-hollow fibre interface, and consequently, a faster response would be expected.

4. Line 64-66: *Although I do agree that fibre surface have higher bias angle than the centre, but a solid fibre can only have centre with physical bias angle. How a hollow fibre can have bias angle, air cannot be physically twisted.*

Response: We thank the reviewer for this comment. The bias angle for the hollow fibre follows the same definition as in the case of solid fibre due to the same kinematic assumptions used in both cases. When subjected to torsion, it was hypothesized that: 1) the radial lines drawn from the center of the cross-section remain straight after the application of twist and 2) adjacent cross-sections that are originally plane and parallel will remain plane and parallel. These are the kinematic assumptions from the elasticity theory for twisting a prismatic bar with cross-section that is axi-symmetric and provide the theoretical base for defining the bias angle. Thus, the definition is applicable to fibre with either solid round or hollow cylindrical-shaped cross-sections since both are axi-symmetric.

5. Line 70: *Twist configured thermoplastic fibres can be heat set above their T_g , to prevent twist loss. This is a well-established method, as used in this paper, so nothing new is to be addressed.*

Response: We thank the reviewer to point out this point. We have removed this

sentence from this paragraph.

6. Line 100-102: There should be some extent of twist loss, when the fibre was released from tethered condition. This might be attributed to the stress relaxation phenomenon of deformed polymer shapes. It is necessary to count those twists loss (even they are small) and consider the number of remaining twists for calculating twist density (as demonstrated in SI Figure 5).

Response: We thank the reviewer to point out this important issue. We counted the twist loss for a different amount of inserted twist after thermal annealing (Supplementary Fig. 12). We added this information in the context, as follows (Page 4, Line 101-104)

After cooling to room temperature (25 °C) and removing the tethering, ~3% twist loss was observed (Supplementary Fig. 12). For the convenience of expression, in the following context, the initially inserted twist was used to calculate the twist density without subtracting the twist loss.

Supplementary Figure 12. The twist loss of the hollow fibre after twist insertion, thermal annealing with both-end tethered at 108 °C for 1h, and cooling down to room

temperature and removing the tethering.

7. Line 103-105: Just inserting twists into a polymer fibre is not the only criteria that give them actuation capability. The fibres should be of highly oriented semicrystalline form. How do the authors know the used commercial hollow fibres have such structure. Experimental evidence is needed.

Response: We thank the reviewer to point out this important issue. We added two-dimensional wide-angle X-ray scattering (2D WAXS) of the PEHF hollow fibre and the results show that the fibres are in highly oriented semicrystalline form (Supplementary Note 1 and Supplementary Fig. 1d). (Page 4, Line 112-115 in the context).

Supplementary Figure 1. (c) The intensity of diffraction peaks versus Bragg angle (2θ) in WAXS for PEHF₅₈₀₋₉₉₀. (d) 2D WAXS diffraction pattern for PEHF₅₈₀₋₉₉₀ showing anisotropic structure. (e, f) The azimuthal curves of (110) and (200) planes in PEHF₅₈₀₋₉₉₀ obtained from (d).

The Herman's orientation parameter (f) is employed to represent the polymer chain alignment of the fibre, where $f = 1$ means that the polymer chains orient

along the fibre axis direction completely; $f = -0.5$ means that the polymer chains orient vertically to the fibre axis direction completely. For a non-oriented sample, f equals to 0.

The Herman's orientation parameters (f_a , f_b , and f_c) for the a-axis, b-axis, and c-axis, for a unit cell of polyethylene crystallite with orthorhombic symmetry, were calculated as 0.12, -0.30, and 0.18, respectively, according to the literatures (*Polym. Eng. Sci.*, 52, 1532, (2012); *J Appl. Polym. Sci.* 18, 1053, (1974)). These values provide the evidences that the fibres are in highly oriented semicrystalline form.

The calculation details are as follows.

The Herman's orientation parameter (f) is defined as:

$$f = \frac{3\langle \cos^2 \Phi \rangle - 1}{2} \quad (1.1),$$

where $\langle \cos^2 \Phi \rangle$ is an orientation factor defined as:

$$\langle \cos^2 \Phi \rangle = \frac{\int_0^\pi I_\Phi \cos^2 \Phi \sin \Phi d\Phi}{\int_0^\pi I_\Phi \sin \Phi d\Phi} \quad (1.2),$$

where I_Φ represents the diffraction intensity. The azimuthal curve obtained by integration of diffraction intensity in the 2D WAXS pattern is used in Eq. (1.2) to calculate $\cos^2 \Phi$ by the numerical integration process. Then the values of $\cos^2 \Phi$ are used in Eq. (1.1) to calculate the orientation function value f of the semicrystalline polymers.

For polyethylene with orthorhombic symmetry of the unit cell, the a-axis (f_a) and b-axis (f_b) orientation parameters were calculated from the intense reflections of (200) and (020) planes based on the relation:

$$\langle \cos^2 \Phi_{110} \rangle = 0.692 \langle \cos^2 \Phi_{020} \rangle + 0.308 \langle \cos^2 \Phi_{200} \rangle \quad (1.3),$$

where $\langle \cos^2 \Phi_{110} \rangle$ and $\langle \cos^2 \Phi_{200} \rangle$ are obtained from the azimuthal intensity measurements on the (110) and (200) reflections, respectively. The parameter f_c is estimated based on the orthogonal symmetry relationship: $f_a + f_b + f_c = 0$. (*J. Polym. Sci.*, 31, 327, (1958))

The detailed calculations for f_a , f_b , and f_c are as follows. First, an azimuthal curve was obtained by integration of diffraction intensity in the 2D WAXS pattern over the azimuthal angle using the software of the GADDS (general area detector

diffraction system) for diffraction angles ($2\theta_d$) of 21.6° (110 plane) and 24° (200 plane), as shown in Supplementary Fig. 1c. Two identical peaks were observed in the azimuthal curve (Supplementary Fig.1d) because of the symmetric diffraction pattern in 2D WAXS. The f_a , f_b , and f_c of the PEHF are respectively 0.12, -0.30, and 0.18, which indicate that the fibres are in highly oriented semicrystalline form.

We added this discussion in the supplementary information (Supplementary Note 1. Page 4-5, Line 89-123).

8. Line 149-158: These are well-known information from many previously published articles. Please remove from the main manuscript, this information better suited in SI.

Response: We thank the reviewer to point out this issue, and we moved this figure to Supplementary Fig. 11 in SI.

9. Line 163-164: How is the response time (1.2 sec) considered? The flow of hot and cold fluid from one end to the other end will certainly have time-dependent temperature gradients. It seems like fluid pass will take more than the response time, please explain. Also, explain the effect of the mass of the weight flown through. Is there any hysteresis of actuation present between cooling and heating cycles? If present, how about using cold (e.g. ice water) fluid to reduce the hysteresis.

Response: We thank the reviewer for these valuable comments. We calculated the time of the fluid passing from one end to the other end at a flowrate of 1.72 g s^{-1} , which is less than 0.05 s. We added an experiment to show the dependence of fluid density on the coil length of the hollow fiber actuator, and the initial actuator length and the actuation performance are negligibly affected by the fluid mass (Supplementary Figs. 4c and d). We added an experiment to show the hysteresis of actuation between cooling and heating cycles for different cooling temperatures, and the results indicated that a lower cooling temperature resulted in lower hysteresis (Supplementary Fig. 5). (Page 14, Line 399-400).

Supplementary Figure 4. (c) The initial coil length and actuation stroke of the PEHF₅₈₀₋₉₉₀ actuator by flowing 25 °C aqueous NaCl solution with different densities. (d) Actuation stroke of the PEHF₅₈₀₋₉₉₀ actuator by flowing 80 °C aqueous NaCl solution with different densities. The spring index was 4.0, and the twist density was 200 turns m⁻¹.

Supplementary Figure 5. The contraction as a function of temperature for the homochiral PEHF₅₈₀₋₉₉₀ actuator driven by consecutively flowing hot water and cold water with a different temperature range. The spring index was 4.0, the inserted twist density was 200 turns m⁻¹, and the flow rate was 1.72 g s⁻¹. The solid symbols represent the heating process, and the open symbols represent the cooling process.

10. Line 169-170: Coiled polymer fibre actuators usually need a few training cycles before having fully reversible actuation cycles (no or insignificant decay). I am not quite convinced that these actuators show no decay from the very first cycle. Please explain the method of how the non-decayed cycles were received from the beginning. Was there any special procedure followed to overcome the creeping effect? The same question also applied for Line 210-213, where effect of different stress on actuation

performance is demonstrated. Repetitive (better to name 'reversible') actuation is certainly possible, but not before training cycles or any novel method applied.

Response: We thank the reviewer for point out this important issue. The hollow-fibre actuators were trained for 3 cycles to obtain repetitive actuation performance. We have added this information in the context and in the experimental section. (Page 4, Line 122-123 in main text; Page 3, Line 69-70 in Supplementary Information)

11. Line 243-245: Why is heat conduction rate along the hollow fibre axis is important? Most of the heat from the hollow core to the polymer sheath might transfer through convection via the fluid. Please explain the significance.

Response: The heat conduction rate along the hollow fiber axis is less important than the convection due to water transport rate and the rate of conduction in the radial direction. We have rewritten this part to make this point clearer, as follows.

Thereafter, we constructed a simple axial temperature model dominated by convection due to water transport since the flow rate of the water is much higher than the axial heat conduction rate of the fibre. (Page 10, Line 285-287)

12. Line 248-250: Please explain the basis of the assumption taken? Is there any relevant reference article?

Response: We thank the reviewer for point out this important issue. The basis for the assumption that the axial heat conduction rate is much lower than the water flow rate was as follows.

First, the high aspect ratio of the hollow fibre makes the heating/cooling process much faster in the radial direction than in the axial direction of the hollow-fibre actuator when considering only heat conduction. Second, polymers are poor heat conductors. When analysing the heat transfer mechanisms in the axial direction of the actuator, convection due to the fast water transport plays a major role compared to the slow heat conduction inside the fibre, which is evident from Supplementary Fig. 3a: the actuation response rate is very sensitive to the water flow rate. We added this discussion in the revised manuscript (Page 10, Line 288-294).

13. Line 261-266: *This model used solid twisted fibres. It is possible that the hollow core twisted fibres follow slightly different torsional actuation mechanisms. If not, please explain the reason.*

Response: We thank the reviewer for this comment. The model stated in line 261-266 is expressed in terms of the dimensions of the outer shell of the fibre and does not necessarily imply that the fiber has to have a solid cross-section. The hollow core twisted fibre follows the same torsional actuation mechanisms as in the case of solid twisted fibre. More specifically, the statement made in lines 261-266 assumes that 1) the radial lines drawn from the center of the cross-section remain straight after the application of twist and 2) adjacent cross sections that are originally plane and parallel will remain plane and parallel. These are the kinematic assumptions from the elasticity theory for twist of a prismatic bar with a cross-section that is axi-symmetric. Thus, the conclusions are applicable to fibres with either solid round or hollow cylindrical shaped cross-sections since both are axi-symmetric.

14. Line 269-270: *Please explain the method of measuring the thermal expansion coefficient of the PE used. Differently drawn semicrystalline polymers will have different coefficients, so simply using reference values will not be feasible for highly sophisticated microfluid manipulation devices. Please include relevant results found from all samples used.*

Response: We added an experiment to measure the thermal expansion coefficients by thermomechanical analysis (TMA). The TMA was performed on the 4 samples that were obtained from the same vendor, and the measured thermal expansion coefficients did not exhibit significant differences. The average axial thermal expansion coefficient (α_λ) between 25 °C and 95 °C was $-(5.3 \pm 0.4) \times 10^{-4} \text{ K}^{-1}$, and the average radial thermal expansion coefficient (α_d) = $(5.2 \pm 0.7) \times 10^{-4} \text{ K}^{-1}$. We updated the data in the context and in the supplementary information. (Page 11, Line 315-317 in the main text; Page 8, Line 204-206 in the supplementary information)

15. Line 278-279: *Is it possible to develop a consolidated equation? Or a step-by-step method? I really don't understand how to correlate the heat transfer equation with the other three reference equations, for theoretically predicting the actuation results.*

Response: We have rewritten the theory part in the main text. We expanded the equation (1) to equations (1a) to (1d), as follows. (Page 9-10, Line 269-284, Page 12, Line 326-329)

To study this transient heat conduction problem, we established a two-step model. First, we modelled the radial heat conduction at an arbitrary cross-section of the hollow-fibre actuator by the following governing equations:

$$\frac{\partial^2 \theta}{\partial r^2} + \frac{1}{r} \frac{\partial \theta}{\partial r} = \frac{1}{\alpha} \frac{\partial \theta}{\partial t} \quad \text{in } R_i \leq r \leq R_o, \quad (1a)$$

$$k \frac{\partial \theta}{\partial r} = h_w (\theta - \theta_w) \quad \text{at } r = R_i, \quad (1b)$$

$$-k \frac{\partial \theta}{\partial r} = h_a (\theta - \theta_a) \quad \text{at } r = R_o, \quad (1c)$$

$$\theta = \theta_0 \quad \text{at } t = 0, \quad (1d)$$

where temperature θ is a function of the radial position r and time t , and the inner and outer radii of the hollow fibre are denoted by R_i and R_o , respectively. Thermal diffusivity $\alpha = k/\rho C_p$ is a material constant that depends on thermal conductivity k , mass density ρ , and specific heat capacity C_p . The convective heat transfer coefficients of water and air are represented by h_w and h_a , respectively. Similarly, θ_w and θ_a are the temperatures of water and air, respectively. Finally, θ_0 denotes the initial temperature of the fibre. Radial temperature $\theta(r, t)$ is obtained by solving the above set of one-dimensional partial differential equations (see Supplementary Note 2 for more details). (Page 9-10, Line 269-284)

The temperature as a function of radial and axial positions and time can be solved using Eqs. (1) and (2). Then, the torsional actuation is obtained from Eq. (4). Finally, the tensile actuation is given by Eq. (5). The temperature solutions to Eqs. (1) and (2) are numerically solved in our work. Although it is possible to derive a closed-form solution, the analytical expression would be very complex (typically in an infinite series form with Bessel functions and some transcendental equations). (Page 12, Line

325-328)

$$\theta_w(x, t) = \theta_{w0} H(vt - x), \quad (2)$$

$$\Delta T = \left(\frac{\Delta \lambda}{\lambda} \frac{1}{\cos^2 \alpha_f} - \frac{\Delta d}{d} \right) T, \quad (3)$$

$$\Delta T = \left(\alpha_\lambda \frac{1}{\cos^2 \alpha_f} - \alpha_d \right) \Delta \theta \cdot T, \quad (4)$$

$$\frac{\Delta L}{L} = \frac{l^2}{NL} \Delta T, \quad (5)$$

16. Line 284-290: Provide appropriate references for these statements.

Response: We added relevant literatures for these statements, as follows.

In high-throughput biosensing or medicine synthesis, a liquid containing specific chemicals is transported to certain vessels for chemical reactions^{78,79}. Because temperature is an important factor influencing reactions, ensuring that the liquid with the required temperature flows into the desired vessel is important for precise control of chemical reactions^{80,81}. However, during microfluid transport, the temperature may vary over time. Therefore, sensing the liquid temperature and sorting the liquid into the desired vessels are highly desirable for microfluidic manipulation.

References:

78. Du G., Fang Q. & den Toonder J. M. Microfluidics for cell-based high throughput screening platforms - A review. *Anal. Chim. Acta.* 903, 36-50 (2016).

79. Wang Z, Kim M. C., Marquez M. & Thorsen T. High-density microfluidic arrays for cell cytotoxicity analysis. *Lab Chip* 7, 740-745 (2007).

80. Hung L. H. & Lee A. P. Microfluidic devices for the synthesis of nanoparticles and biomaterials. *J Med. Biol. Eng.* 27, 1 (2007).

81. Brivio M., Verboom W. & Reinhoudt D. N. Miniaturized continuous flow reaction vessels: influence on chemical reactions. *Lab Chip* 6, 329-344 (2006).

17. Line 305-306: What does it mean by 'linear increase in the rotation angle'? Fig. 5d should be modified based on the explanation provided, why the angle goes from negative to positive in the current figure is not understandable.

Response: The negative value of the rotation angle was because we are using the configuration at 25°C as the reference. In the revised Fig. 5d, we have shifted the reference configuration to 22°C, and as a result, the rotation angle starts from 0° to a positive value. This sentence was changed to: “The rotation angle linearly increased with an increase in the fluid temperature.

Fig. 5 (d) Rotation angle as a function of water temperature.

18. Line 387-390: How the ‘isobaric stresses’ can be explained, when the coiled fibre diameter will change during contraction and expansion? Was there the dynamic change of pressure (stress) considered? Or the helical hollow fibre diameter was used, which does not make sense for such a vertically suspended system.

Response: We use the same expression method according to the previously published papers (*Science* 338, 928 (2012), *Science* 343, 868 (2014), and *Mater. Horiz.* 7, 3305-3315 (2020)), where the fiber cross-sectional area was used to calculate the isobaric stress. So, in this study, the isobaric stress was calculated as the weight of the load divided by the cross-sectional area of the sheath of the hollow fiber. We added the description in the context to make it clear (Page 8, Line 235-237). For better understanding, we also provided the figures directly using the mass of the load in place of the stress in the main context and supporting information (Figs. 4b, 4c and Supplementary Fig. 8).

Response to the reviewer #3:

This paper presents fast torsional and tensile actuators made of hollow fibers, which can detect the fluid temperature and sort the fluid into the desired vessels. It is interesting and well written. Experiments are well planned and conducted. Analysis and conclusions are sound. It shows great application prospects. Some minor problems should be addressed by the authors as follows:

Response: We thank the reviewer for these insightful comments, and we have made substantial revision to improve this manuscript.

1. *“The fluid-driven actuation was 27 times as that driven by air flow; the work capacity and power density were 1.5 times and 90 times, respectively”. This reviewer can see where the data supporting the statement from Fig. 3F and fig. 4F. Can the authors move the supplementary information in the main text as this is an important statement.*

Response: This is a good suggestion. We updated Figs. 1 and 3. The updated Fig. 1c supports the data in Fig. 3f, and Fig. 4e supports the data in Fig. 4f.

2. *“To facilitate counting of the rotation angle, a paddle (with a mass of 1/50 of the hollow fiber) was taped at the outlet of the PEHF580-990 (Fig. 2b)”. Please indicate the paddle by a clear method as the reviewer cannot find it in Fig. 2b.*

Response: This is a good suggestion. We indicated the paddle in Fig. 2b and showed a magnified image to show the paddle.

3. *In Supporting information video1: On the upper left part, leakage occurs and water jet out of the actuator (not the droplets). It seems that the actuator has a pole. Please explain this phenomenon.*

Response: Thank you for pointing out this issue. We re-took a video for this demonstration, and no leakage occurred in the updated video.

4. *The fluid in this article is mainly water, but there is no discussion on other fluids.*

There are many kinds of fluids, some of which are more viscous and not suitable for such a small-diameter pipe. Please explain the range of practical fluids at the end of the summary, or summarize the prospect.

Response: We thank the reviewer for this good suggestion. We added experiments to show the actuation dependence on the type of the applied fluid, fluid viscosity, and fluid density, and there is a negligible change in the actuation performance when actuated by the different investigated types of fluid (Supplementary Figs. 2d and 4). We added this information in the discussions (Page 14, Line 396-399)

Supplementary Figure 2d. (d) The torsional actuation of the PEHF₅₈₀₋₉₉₀ actuator by flowing different types of 40 °C liquid at a flow rate of 1.72 g s⁻¹. The room temperature is 25 °C, the environmental relative humidity is 40%, and the water flow rate is 1.72 g s⁻¹. EtOH: ethanol; EA: ethyl acetate; DMSO: dimethyl sulfoxide; THF: tetrahydrofuran; DMF: N, N-dimethylformamide; DCM: dichloromethane; IPA: iso-propanol; PE: petroleum ether.

Supplementary Figure 4. (a) The initial coil length and actuation stroke of the homochiral PEHF₅₈₀₋₉₉₀ actuator by flowing 25 °C glycerol/water solution with different viscosity. (b) Actuation stroke of the PEHF₅₈₀₋₉₉₀ actuator by flowing 80 °C glycerol/water solution with different viscosity. (c) The initial coil length and actuation stroke of the PEHF₅₈₀₋₉₉₀ actuator by flowing 25 °C aqueous NaCl solution with different densities. (d) Actuation stroke of the PEHF₅₈₀₋₉₉₀ actuator by flowing 80 °C aqueous NaCl solution with different densities. The spring index was 4.0, and the twist density was 200 turns m⁻¹.

5. Supplementary Figure 1. (a). It is better to delete the data at the end of this curve (the vertical part). Because the data of this part were measured when the fiber was cracked, thus the slop is nearly ∞ .

Response: We updated the Supplementary Fig. 1a by deleting the data at the end of this curve.

Supplementary Figure 1. (a) The stress-strain curve of the PEHF₅₈₀₋₉₉₀.

6. “In recent years, stimuli-responsive materials have been incorporated into micrometer channels to realize microfluidic contro” (missing letter ‘l’, please check the whole paper that there is no cacography!)

Response: Thank you for your careful reading. We re-checked the typos and grammar carefully and also have them checked by an English language service.

This document certifies that the manuscript
Microfluidic Manipulation by Spiral Hollow-Fiber Actuators

prepared by the authors

Sitong Li, Rui Zhang, Guanghao Zhang, Luyizheng Shuai, Wang Chang, Xiaoyu Hu, Min Zou, Xiang Zhou, Baigang An, Dong Qian, Zunfeng Liu*

was edited for proper English language, grammar, punctuation, spelling, and overall style
by one or more of the highly qualified native English speaking editors at SNAS.

This certificate was issued on **December 7, 2021** and may be verified
on the SNAS website using the verification code **E505-C78B-1FOC-CF9E-D36P**.

Neither the research content nor the authors' intentions were altered in any way during the editing process. Documents receiving this certification should be English-ready for publication; however, the author has the ability to accept or reject our suggestions and changes. To verify the final

SNAS edited version, please visit our verification page at secure.authorservices.springernature.com/certificate/verify.

If you have any questions or concerns about this edited document, please contact SNAS at support@as.springernature.com.

REVIEWERS' COMMENTS

Reviewer #1 (Remarks to the Author):

The authors responded well to the comments.

Reviewer #2 (Remarks to the Author):

The authors now have demonstrated significant improvement in the paper. This includes the understanding and representation of true actuation mechanism, and factors that fundamentally influence the demonstrated system. This paper can now be recommended for publication.

Reviewer #3 (Remarks to the Author):

The authors have revised the manuscript and addressed my questions satisfactorily.

- A separate point-by-point response to the reviewers' comments, reproduced verbatim.

Reply: Because there is no further comments from the reviewers, we thank all the reviewers for their efforts for improving this manuscript.

Response to the reviewer #1:

The authors responded well to the comments.

Response:

We thank the reviewer for this comment.

Response to the reviewer #2:

The authors now have demonstrated significant improvement in the paper. This includes the understanding and representation of true actuation mechanism, and factors that fundamentally influence the demonstrated system. This paper can now be recommended for publication.

Response: We thank the reviewer for this comment.

Response to the reviewer #3:

The authors have revised the manuscript and addressed my questions satisfactorily.

Response: We thank the reviewer for this comment.